# Precision stratification of prognostic risk factors associated with outcomes in gestational diabetes mellitus: a systematic review

Zhila Semnani-Azad[1✉], Romy Gaillard[2], Alice E. Hughes[3], Kristen E. Boyle [4], Deirdre K. Tobias[1,5] ADA/EASD PMDI* & Wei Perng [6]

### Abstract

**Background** The objective of this systematic review is to identify prognostic factors among women and their offspring affected by gestational diabetes mellitus (GDM), focusing on endpoints of cardiovascular disease (CVD) and type 2 diabetes (T2D) for women, and cardiometabolic profile for offspring.

**Methods** This review included studies published in English language from January 1st, 1990, through September 30th, 2021, that focused on the above outcomes of interest with respect to sociodemographic factors, lifestyle and behavioral characteristics, traditional clinical traits, and 'omics biomarkers in the mothers and offspring during the perinatal/postpartum periods and across the lifecourse. Studies that did not report associations of prognostic factors with outcomes of interest among GDM-exposed women or children were excluded.

**Results** Here, we identified 109 publications comprising 98 observational studies and 11 randomized-controlled trials. Findings indicate that GDM severity, maternal obesity, race/ethnicity, and unhealthy diet and physical activity levels predict T2D and CVD in women, and greater cardiometabolic risk in offspring. However, using the Diabetes Canada 2018 Clinical Practice Guidelines for studies, the level of evidence was low due to potential for confounding, reverse causation, and selection biases.

**Conclusions** GDM pregnancies with greater severity, as well as those accompanied by maternal obesity, unhealthy diet, and low physical activity, as well as cases that occur among women who identify as racial/ethnic minorities are associated with worse cardiometabolic prognosis in mothers and offspring. However, given the low quality of evidence, prospective studies with detailed covariate data collection and high fidelity of follow-up are warranted.

### Plain language summary

Gestational diabetes mellitus (GDM) occurs when levels of sugar in the blood are high during pregnancy. We sought to identify factors associated with short- and long-term cardiometabolic disease risk, health conditions that involve heart-related issues and complications in bodily function, among women with GDM and their offspring. We reviewed publications on factors related to type 2 diabetes (T2D) and cardiovascular disease (CVD) risk among women with GDM, and additionally assessed body composition in offspring of women with GDM. We found that GDM severity, maternal obesity, self-identified race/ethnicity, poor diet, and low physical activity levels predict postpartum T2D and CVD in the women, and unfavorable long-term cardiometabolic disease risk in offspring. The quality of evidence was poor, emphasizing a need for high-quality research capturing detailed short- and long-term outcome data to facilitate preventative interventions to improve health of women and children.

[1] Department of Nutrition, Harvard T.H. Chan School of Public Health, Boston, MA, USA. [2] Department of Pediatrics, Erasmus MC, University Medical Center, Rotterdam, the Netherlands. [3] Faculty of Health and Life Sciences, University of Exeter Medical School, Exeter, UK. [4] Department of Pediatrics and the Lifecourse Epidemiology of Adiposity and Diabetes (LEAD) Center, University of Colorado Anschutz Medical Campus, Aurora, CO, USA. [5] Department of Medicine, Brigham and Women's Hospital, Harvard Medical School, Boston, MA, USA. [6] Department of Epidemiology and the Lifecourse Epidemiology of Adiposity and Diabetes (LEAD) Center, University of Colorado Anschutz Medical Campus, Aurora, CO, USA. *A list of authors and their affiliations appears at the end of the paper. ✉email: zsemnaniazad@hsph.harvard.edu

Gestational diabetes mellitus (GDM), a state of hyperglycemia due to insufficient insulin secretion and/or insulin resistance that occurs during pregnancy, is the most common metabolic disorder of pregnancy, affecting 6–12% of pregnancies globally[1,2]. A diagnosis of GDM is not only associated with risk of acute pregnancy and delivery complications, but also carries implications for the long-term risk of type 2 diabetes (T2D)[3,4] and cardiovascular disease (CVD)[5]. Additionally, offspring exposed to GDM in utero have higher adiposity and a worse metabolic profile across the life course than their unexposed counterparts[6,7]. The wide-ranging and intergenerational sequelae of GDM-affected pregnancies emphasize the importance of characterizing not only the short- and long-term consequences of this common pregnancy complication. Further, identification of bellwethers of such consequences will facilitate preventive intervention of such comorbidities and complications, also known as disease prognosis.

Recent technological advancements have improved the capacity to comprehensively assess physiology. In turn, these developments facilitated the ability to harness metabolic heterogeneity – the phenomenon of interest to precision medicine by which similar exposures and risk factors yield differential health sequelae across individuals. In the context of GDM prognosis, this effort requires the identification of prognostic factors and biomarkers among women with a history of GDM and/or their offspring who were exposed to GDM in utero that may serve as both causal and non-causal indicators of future health risks.

Recognizing the relevance of metabolic heterogeneity in accurate and precise assessment of disease prediction, diagnosis, treatment, and prognosis, the Precision Medicine in Diabetes Initiative (PMDI) was established in 2018 by the American Diabetes Association (ADA) in partnership with the European Association for the Study of Diabetes (EASD). The ADA/EASD PMDI includes global thought leaders in precision diabetes medicine who are working to address the burgeoning need for better diabetes prevention and care through precision medicine[8]. This Systematic Review is written on behalf of the ADA/EASD PMDI as part of a comprehensive evidence evaluation in support of the 2nd International Consensus Report on Precision Diabetes Medicine[9].

Thus, in an effort to evaluate prognostic factors to better understand health risks related to postpartum and long-term cardiometabolic health outcomes among mothers with GDM and her offspring, we conducted a systematic review that synthesizes evidence from empirical research papers published through September 1st, 2021, to evaluate and identify prognostic conditions, risk factors, and biomarkers among women and offspring affected by GDM pregnancies, focusing on clinical endpoints of CVD and T2D among women with a history of GDM; and adiposity and cardiometabolic risk profile among offspring exposed to GDM in utero. Overall, we find that GDM severity, maternal obesity, self-identified race/ethnicity, poor diet, and low physical activity levels predict postpartum T2D and CVD in the women, and unfavorable long-term cardiometabolic health in offspring with GDM exposure.

## Methods

**Systematic review protocol development**. We registered our search strategy and systematic review protocol to PROSPERO CRD42021276094[10]. We developed a systematic review protocol to comprehensively include and evaluate individual research studies reporting on risk factors for long-term clinical outcomes in women with GDM and a range of cardiometabolic health and anthropometric outcomes in GDM-exposed offspring. Nota bene, ADA/EASD PDMI is committed to using inclusive language,

especially in relation to gender. We choose to use gendered terminology throughout the article following the rationale for using gendered language in studies of maternal and child health, including but not limited to reducing risk of exposure misclassification and avoidance of dehumanizing terms[11]. Further, most of the original studies reviewed used 'women' as their terminology to describe their population, as GDM per definition occurs in pregnancy which can only occurs in individuals that are female at birth. In this review, we use the term 'women' throughout, but acknowledge that not all individuals who experienced a pregnancy may self-identify as a woman.

Our strategy aimed to identify two broad categories of empirical studies: (1) populations of women with a history of prior GDM that investigated additional exposures or risk factors for incident postpartum T2D or CVD; (2) populations comprising offspring exposed to GDM in utero that investigated additional exposures or risk factors for an adverse cardiometabolic profile. Studies including pregnancies unaffected by GDM were eligible only if results were included for GDM subgroups.

Prognostic factors of interest, hereafter also referred to as exposures, included sociodemographic factors, lifestyle and behavioral characteristics, traditional clinical traits, and 'omics biomarkers. We considered these prognostic factors during the perinatal/postpartum periods and across the lifecourse for both the mothers and offspring. Maternal outcomes of interest were incident T2D or CVD, including study-specific composites of clinical cardiovascular events, non-fatal and fatal myocardial infarction or stroke, and chronic kidney disease (CKD). For offspring, we were interested in outcomes reported 12 weeks of age and older, and limited to anthropometrics, glycemic and cardiometabolic traits or biomarkers, and incident metabolic syndrome (MetS), T2D, or CVD.

**Data sources, search strategy, and screening criteria**. We developed search terms for Medline EMBASE, and Cochrane CENTRAL (Supplementary Data 1) for eligible citations published in English language from January 1st, 1990, through September 30th, 2021. References of accepted manuscripts and relevant systematic reviews published within the past 2 years were screened to identify additional citations. We included prospective and retrospective observational studies identifying factors with incident outcomes of interest in women or offspring exposed to GDM. We excluded cross-sectional analyses among populations with prevalent disease outcomes or traits. While studies could include non-GDM exposed pregnancies, those without subgroup findings exclusively among GDM pregnancies were excluded. We also included interventions prospectively comparing effects of a treatment assignment on the outcome. Exclusion criteria comprised studies with outcomes <6 weeks postpartum, maternal studies reporting only intermediate phenotypes, glycemic traits, or cardiometabolic biomarkers, and studies in offspring that only assessed endpoints outside of the cardiometabolic outcomes of interest (e.g., neurodevelopment, allergic disease). Using these, two independent reviewers conducted screening at the title abstract level. For accepted citations, two independent reviewers implemented screening of the full manuscripts. Conflicts at all screening stages were resolved by a third reviewer. All screening was conducted in the Covidence online systematic review tracking platform.

**Data extraction and synthesis of results**. We developed and piloted a data extraction template for eligible manuscripts. Data included manuscript information, study level details and design, population enrollment and characteristics, exposure and outcome ascertainment and diagnosis criteria, follow-up time of outcome

assessment since index GDM pregnancy and other pertinent details. We indicated the population in which outcomes were assessed (e.g., maternal, offspring, or both), and recorded the exposures that were investigated in four broad categories: (i) social/genetics factors across the life course; (ii) all factors in perinatal/postpartum window; (iii) long-term maternal exposures; and (iv) long-term offspring exposures.

**Quality assessment (risk of bias) and synthesis**. We assessed the quality of each study using the Joanna Briggs Institute's (JBI) critical appraisal tools for cohort studies and randomized controlled trials (RCTs)[9]. For cohort studies, we assessed quality based on 11 items which evaluated population recruitment, exposure and outcome ascertainment, confounding, statistical methodology, and follow-up. For the RCTs, the JBI criteria evaluated 13 items which assessed selection and allocation, intervention, administration, outcome ascertainment, follow-up, and statistical analysis. Each JBI item was categorized as, 'Yes,' 'No,' 'Unclear,' or 'Not applicable' following the guidelines. Any uncertainty in assessment was further discussed by the full research team.

**Overall evidence certainty assessment and synthesis**. The certainty of evidence was determined using the Diabetes Canada 2018 Clinical Practice Guidelines for studies[12]. Levels were based on study design and criteria focused on inception cohort of patients presenting GDM but without outcomes of interest, inclusion/exclusion reproducibility, follow-up of at least 80% of participants and assessment of loss to follow-up, adjustment for confounding factors, and reproducible outcome measures. Scoring ranged from level 1 to 4, with Level 1 indicating the highest certainty of evidence and Level 4 indicating the lowest certainty of evidence. Details on the criteria and guidelines are in Supplementary Table 1.

**Reporting summary**. Further information on research design is available in the Nature Portfolio Reporting Summary linked to this article.

## Results
Of the 8141 studies identified, five were excluded due to duplication (Fig. 1). Another 7770 were excluded following title and abstract review. The remaining 366 studies were reviewed in full, of which 106 studies met the inclusion criteria through the database search. An additional three studies were identified through manual search. A total of 109 studies were included in this review.

Of the 109 included, 98 were observational studies and 11 were RCTs (Supplementary Data 2 and 3). Of the studies, 51 focused on maternal outcomes and 38 focused on offspring outcomes. Of the RCTs, three evaluated maternal outcomes and eight assessed offspring outcomes. Studies included data from primarily from white populations from North America and Europe. Sample sizes of the eligible studies ranged from 26 to 23,302.

### Maternal outcomes
*Maternal type 2 diabetes*. Forty-nine observational studies (Supplementary Data 4) and two RCTs (Supplementary Data 5) assessed sociodemographic, lifestyle, clinical, and pregnancy characteristics associated with the risk of T2D among GDM women. The most frequently studied characteristics were maternal BMI and GDM severity. All observational studies that assessed maternal BMI as a prognostic factor showed that higher maternal BMI prior to and/or during pregnancy, and later in the lifecourse predicted higher risk of T2D. One observational

study[13] further demonstrated that a greater pre-pregnancy weight increased the risk of T2D, though this study did not observe a significant association of gestational weight gain with T2D (Supplementary Data 4). Seventeen observational studies, including one that derived a composite risk score for future T2D risk[14], assessed GDM severity in relation to risk of T2D. Findings indicate that more severe GDM, measured by either clinical markers assessing degree of hyperglycemia or need for insulin treatment, predicts risk of developing T2D (Supplementary Data 4). Fewer studies examined the role of lifestyle behaviors and prenatal clinical characteristics. Four observational studies[15–18] investigated the role of self-identified race/ethnicity – which we view as social constructs as opposed to biological forms of determinism – for the risk of T2D, two of which showed no significant associations[15,17] and two suggested that the risk was higher among women with non-white European ancestry[16,18] (Supplementary Data 4).

Four[19–22] of seven[17,19–24] observational studies that focused on prognostic value of pregnancy or delivery complications reported that additional pregnancy complications beyond GDM conferred higher risk of T2D. The pregnancy complications assessed varied across reports including stillbirth, gestational hypertension, and cesarian section. Seven studies explored the role of parity[24–30], of which five[25–28,30] found that higher parity predicted risk of T2D. Four observational studies[31–34] showed that breastfeeding was associated with a reduced risk of developing T2D in later life. Two observational studies[35,36] and one RCT[37] assessed associations of healthy dietary patterns during mid-life with risk of incident T2D among women with a history of GDM but showed inconsistent results. Ten studies assessed biomarkers of T2D risk[14,30,38–45], including metabolomics, lipidomics, sICAM and sE-selectin, and proinsulin-to-insulin ratio.

**Maternal cardiovascular diseases**. Six observational studies[19,46–50] explored the role of sociodemographic, lifestyle, and pregnancy characteristics in future risk of CVD among women with GDM (Supplementary Data 6). Two studies identified maternal BMI before[46] and during[48] pregnancy as risk factors for future CVD, in which women with overweight or obesity, in addition to GDM, have a higher risk of CVD as compared to normal weight women with GDM. One study[47] further showed that a healthy lifestyle – i.e., healthy diet, physical activity, and being a non-smoker – was associated with a lower risk of CVD. Two studies showed that pregnancy complications— namely, gestational hypertension[50] and stillbirth[19]—predicted risk of CVD. No effect modification was identified with respect to family history of CVD[47] or chronic hypertension[48].

**Quality of studies conducted and certainty of evidence in women with a history of GDM**. The quality of studies for prognostic factors indicative of future T2D or CVD risk is low and the overall certainty of evidence ranked between Levels 3 and 4 according to the Diabetes Canada 2018 Clinical Practice Guidelines[12]. (Fig. 2 for observational studies; Fig. 3 for RCTs). Most current literature were based on retrospective studies leveraging registry data and observational cohort studies, both of which are vulnerable to bias due to residual confounding, reverse causation bias by pre-existing conditions, and other characteristics around the time of pregnancy and GDM diagnoses.

### Offspring outcomes
*Anthropometry and body composition*. In comparison to the large maternal literature, relatively few studies focused on prognostic factors associated with suboptimal offspring body composition among those exposed to GDM *in utero*. Forty observational studies

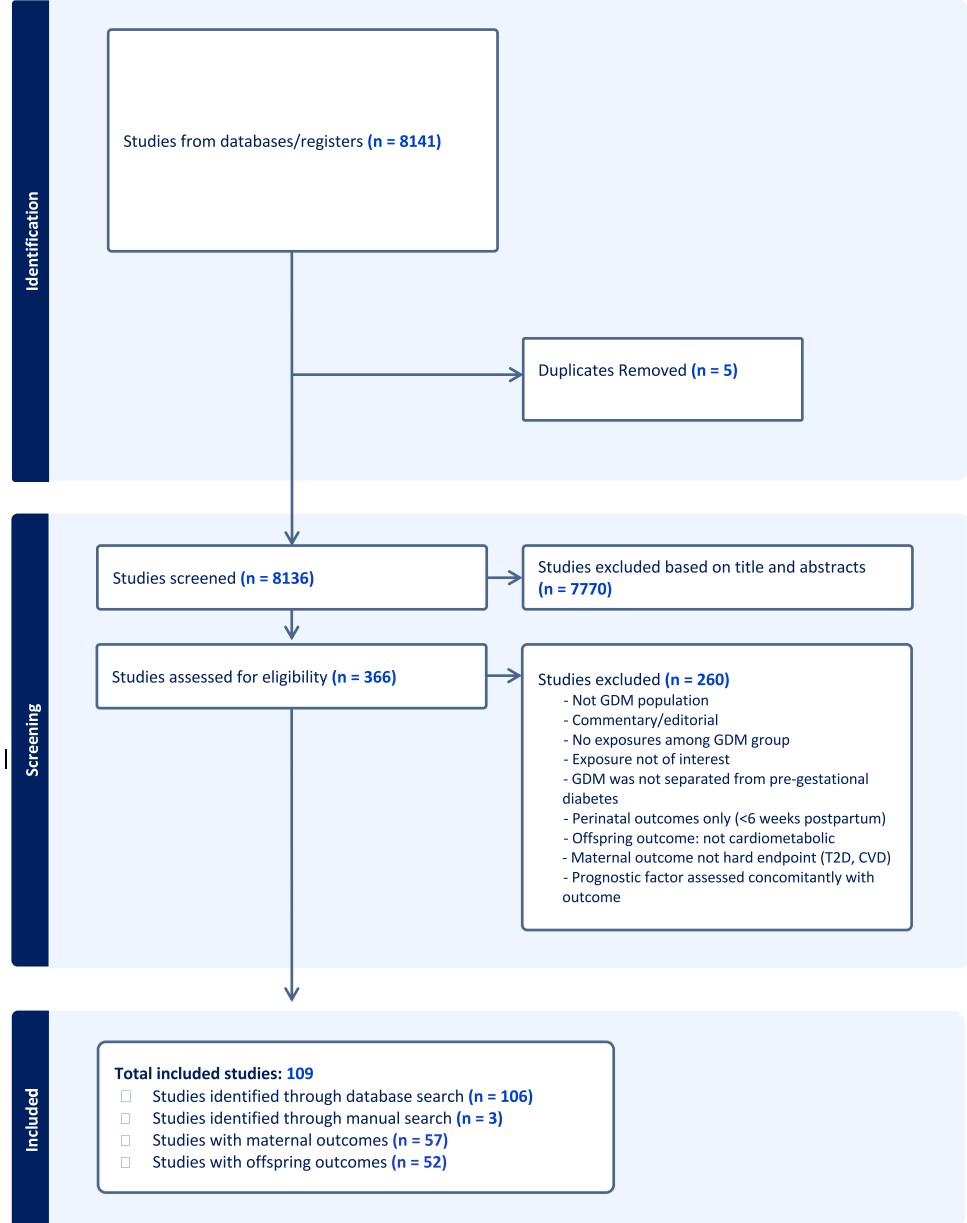

The PRISMA flow diagram details the search and selection process applied in the review.

**Fig. 1** Preferred Reporting Items for Systematic Reviews and Meta-Analyses (PRISMA) flow diagrams for study identification, screening, and retention of studies included in this systematic review.

(Supplementary Data 7) and five RCTs (Supplementary Data 8) examined associations of sociodemographic, lifestyle, clinical and pregnancy characteristics associated with anthropometric outcomes in offspring of GDM women. The RCTs, by nature, also enabled assessment of the effect of GDM treatment type (e.g., Metformin vs. insulin; dietary advice, glucose monitoring, and insulin therapy vs. routine care) on offspring outcomes.

The most studied associations included maternal BMI, GDM severity, breastfeeding status, and offspring birthweight, in relation to offspring anthropometric outcomes (e.g., BMI and risk of overweight/obesity). Seven observational studies[51–57] found that higher maternal pre-pregnancy BMI was associated with higher adiposity in the offspring, as reflected by a higher BMI, waist circumference or directly-assessed fat mass, and greater risk of overweight or obesity. Nine studies[55,56,58–64] assessed the associations of maternal GDM severity, measured by either clinical markers of hyperglycemia or need for insulin

treatment, with offspring body composition, of which four observational studies[55,56,63,64] indicated that more severe maternal GDM is associated with a higher offspring BMI and overweight risk. RCTs that evaluated GDM severity and showed no significant association with offspring anthropometry or body composition.

Six[55–57,60,65,66] of 10 observational studies[51,55–57,60,65–69] showed that a larger size and/or higher adiposity at birth predicts higher future BMI and risk of overweight among GDM-exposed offspring.

With regards to breastfeeding status, one study[69] reported that breastfed offspring with larger size at birth had lower future BMI and lower risk of overweight or obesity. Multiple observational studies showed that exclusive breastfeeding and longer vs. shorter duration of breastfeeding are associated with lower offspring BMI and risk of overweight or obesity (Supplementary Data 7). Additionally, a study in the SWIFT

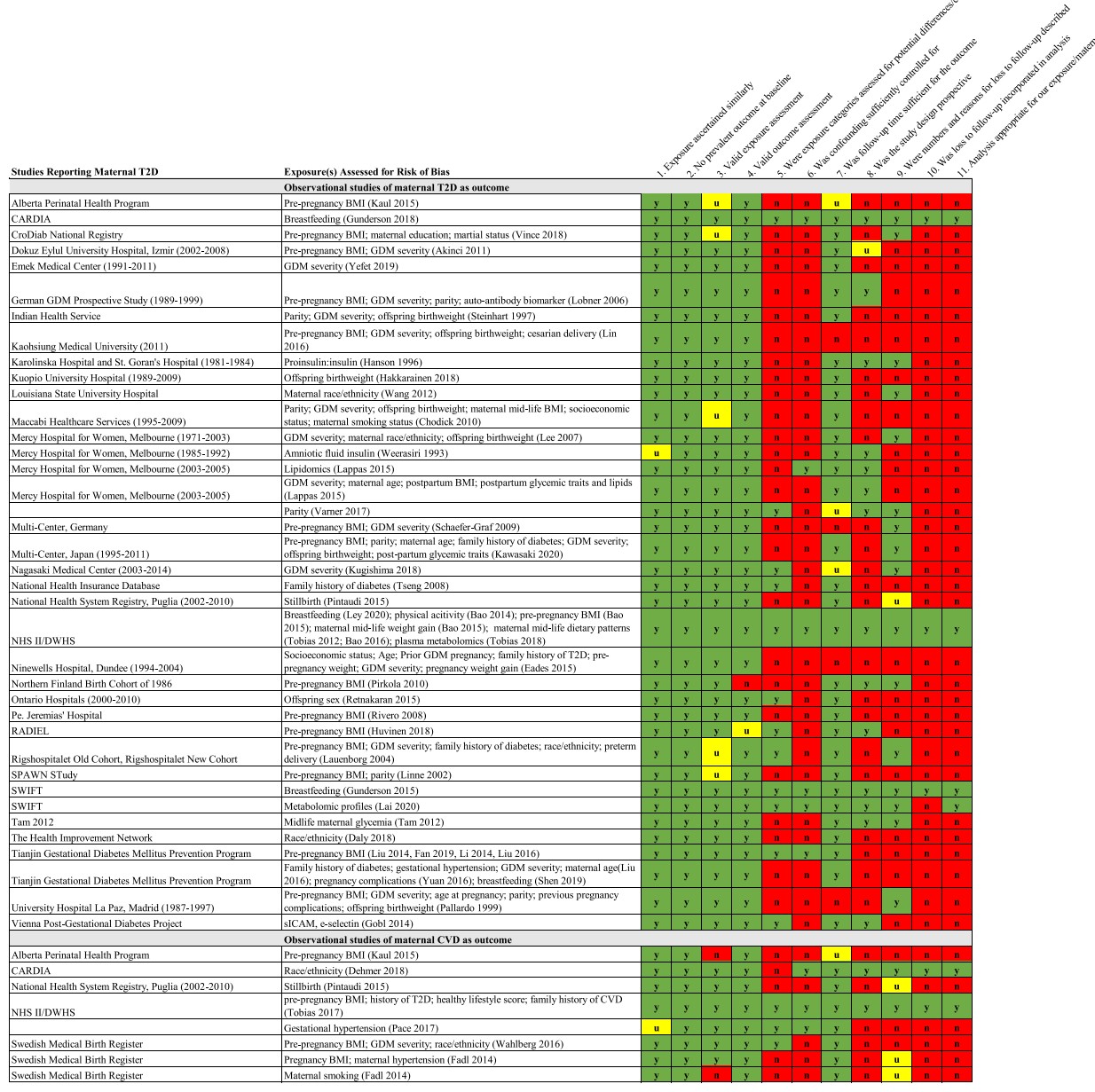

**Fig. 2 Heat map of study quality according to the Diabetes Canada Clinical Practice Guidelines for observational studies assess maternal type 2 diabetes (T2D) and cardiovascular disease (CVD) as outcomes.** Green cells indicate high quality; red cells indicate low quality. Yellow cells indicate unclear/unable to assess quality based on information provided.

cohort showed that inadequate duration and/or exclusivity of breastfeeding, alone and in combination with consumption of fruit juice or sugar sweetened beverages during the first year of life, predicts higher offspring BMI at ages 2–5 years[70]. Three studies using data from the Danish National Birth Cohort indicated that maternal prenatal diet consisting of fatty fish[71], refined grain[72], and sugar-sweetened beverage intake[73] were associated with higher offspring BMI, whereas protein intake[74] and glycemic index/load[75] did not show significant impact on offspring abdominal fat. Finally, one study identified a genetic risk score that predicted higher BMI among offspring exposed to GDM in utero[76].

Of the five RCTs testing an effect of GDM treatment on offspring anthropometry and body composition, three[77–79] yielded

null findings and two found that treatment with Metformin, as compared to insulin, was associated with higher offspring adiposity according to skinfold thicknesses[80] and weight[81] within the first 18 months of life (Supplementary Data 8).

**Cardiometabolic profile.** We identified fourteen observational studies (Supplementary Data 9) and five RCTs (Supplementary Data 10) that evaluated prognostic risk factors for adverse cardiometabolic outcomes among GDM-exposed offspring. These studies focused on blood pressure, lipids, and glycemic markers in the offspring separately or via a score comprising multiple biomarkers.

Birthweight was the most studied predictor of the offspring prognostic factors, but only two observational studies[82,83] showed that a higher birthweight predicted MetS components in offspring

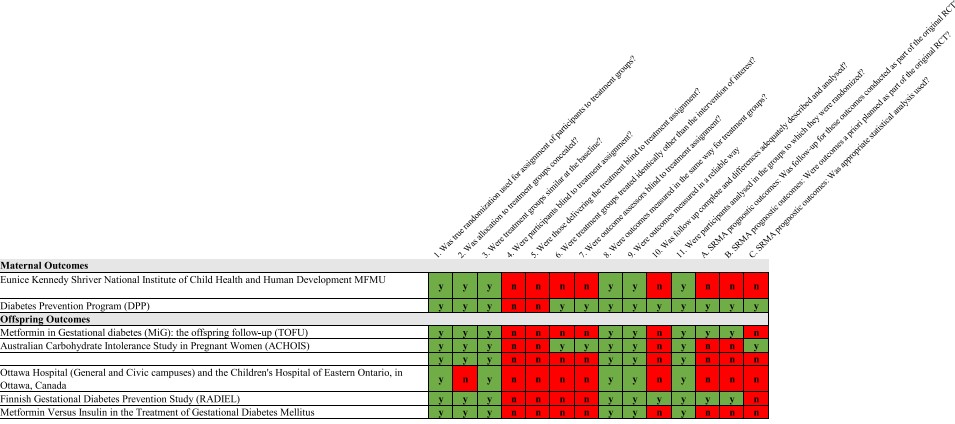

**Fig. 3 Heat map of study quality according to the Diabetes Canada Clinical Practice Guidelines for randomized controlled trials (RCTs) assessing GDM intervention on maternal and offspring outcomes.** Green cells indicate high quality; red cells indicate low quality. Yellow cells indicate unclear/unable to assess quality based on information provided.

later in life. Four observational studies assessed associations of specific maternal dietary components (glycemic index/load[75], fish[71], magnesium[84], and protein[74]), though no consistent associations were observed in relation to offspring cardiometabolic outcomes. Although one observational study showed that breastfeeding was associated with a lower risk of a MetS phenotype in the offspring[85], but this finding was not recapitulated in other observational studies.

Several RCTs compared diet vs. insulin treatment of GDM and showed no significant associations with the development of a MetS phenotype in the offspring (Supplementary Data 10). One RCT[86] assessed the effect of a lifestyle intervention comprising exercise and diet counselling for treatment of GDM vs. usual clinical care and found higher risk of unfavorable metabolic outcomes among offspring in the intervention group.

**Quality of studies and certainty of evidence conducted in offspring exposed to GDM in utero.** We identified low quality of evidence for prognostic factors indicative of future adiposity and cardiometabolic risk among offspring exposed to GDM *in utero* (Fig. 3 for RCTs; Fig. 4 for observational studies). As with the maternal literature, most studies focusing on offspring outcomes were based on retrospective study designs leveraging registry data and observational cohort studies, both of which can be fraught with residual confounding and reverse causation bias, as well as structural biases like selection and attrition bias. Moreover, the literature of offspring outcomes remains scant and with potentially inadequate durations of follow-up for manifestation of clinically relevant cardiometabolic outcomes, though additional research is warranted. Furthermore, the certainty of evidence for maternal and offspring exposures with cardiometabolic outcomes were scored at Level 4[12], based on several factors including limited studies, small sample sizes, heterogeneity of study designs, and inadequate statistical methods.

## Discussion
**Summary**. This systematic review sought to identify prognostic risk factors during the perinatal period and across the lifecourse for maternal and offspring cardiovascular and metabolic outcomes among women and offspring affected by GDM pregnancies. We hypothesized that worse glycemic control at the time of GDM diagnosis (i.e., severity of GDM), older maternal age, belonging to a racial/ethnic minority group as proxy of upstream social experiences that trickle down to affect physiology[87], unhealthy lifestyle behaviors during the prenatal period (i.e., poor diet quality and low physical activity levels) predict risk of

incident type 2 diabetes (T2D) and cardiovascular disease (CVD) among women with a history of GDM, and an unfavorable cardiometabolic profile among offspring exposed to GDM in utero.

The studies identified herein were primarily long-term retrospective and prospective studies. The level of evidence for prognostic risk factors of maternal T2D and CVD and for offspring cardiometabolic risk is low due to unmeasured confounding by lifestyle behaviors, the possibility of reverse causation bias due to pre-existing chronic conditions prior to or at the time of GDM diagnosis. Additionally, for offspring outcomes, the small body of literature on prognostic factors indicative of future adiposity and cardiometabolic risk and major loss to follow-up in both observational and intervention studies.

**Maternal outcomes**. Among women with GDM, higher BMI at any time in relation to the index pregnancy – i.e., pre-pregnancy, during the index pregnancy including gestational weight gain, and lifecourse measures of weight – predicted higher risk of T2D later in life. GDM severity, typically estimated by use of insulin or higher blood glucose values during the index pregnancy, was consistently associated with higher risk of developing T2D. While few studies assessed race and/or ethnicity as a prognostic risk factor, women of Asian or non-white European descent with a history of GDM had higher risk of future T2D than white women[16,18,46]. Breastfeeding duration and/or exclusivity was consistently associated with lower risk T2D risk following a GDM diagnosis during pregnancy, though follow-up often ended <2 years postpartum—a period within which occult T2D incidence is relatively low (Supplementary Data 4). Longer duration follow-up is necessary to better evaluate the benefits of breastfeeding on T2D risk. Some observational studies indicated a protective effect of lifestyle factors such as physical activity level during the perinatal and postpartum periods, and compliance with a healthy diet (e.g., adherence to a Mediterranean or DASH-like dietary pattern; the Healthy Eating Index score). However, RCTs investigating the effects of dietary interventions yielded mixed results (Supplementary Data 5). Several observational studies also examined biomarkers of T2D risk following GDM pregnancies, including degree of hyperglycemia at the time of GDM diagnosis, lipids, inflammation, and metabolomics biomarkers[38–40]. However, low certainty of evidence from the studies and lack of replication/validation of findings prevent us from drawing firm conclusions regarding which factors may be the best predictors of future diabetes.

In line with a large literature demonstrating that women with a history of GDM are at higher risk CVD than their non-diabetic

| Studies Reporting Offspring Outcomes | Exposure(s) Assessed for Risk of Bias | 1. Exposure ascertained similarly | 2. No prevalent outcome at baseline | 3. Valid exposure assessment | 4. Valid outcome assessment | 5. Were exposure categories assessed for potential differences/confounders | 6. Was confounding sufficiently controlled for | 7. Was follow-up time sufficient for the outcome | 8. Was the study design prospective | 9. Were numbers and reasons for loss to follow-up described | 10. Was loss to follow-up incorporated in analysis | 11. Analysis appropriate for our exposure/maternal outcome |
|---|---|---|---|---|---|---|---|---|---|---|---|---|
| University Medical Center Utrecht (1990-2006) | Large for gestational age (Hammoud 2018) | y | y | y | y | y | n | y | n | n | n | n |
| Ahvazi Health Care Centers (2014-2016) | Offspring sex (Nouhjah 2019) | y | y | y | y | n | n | y | n | n | n | n |
| Calgary Region Births (2005-2013) | Breastfeeding; large for gestational age (Kaul 2019) | y | y | y | y | y | n | y | n | n | n | n |
| Children of 1997 Cohort | Breastfeeding (Hui 2018) | y | y | y | y | y | y | y | n | n | y | n |
| Clinical Hospital of Obstetrics and Gynecology, Poznan | GDM severity (Wroblewska-Seniuk 2009) | y | y | y | y | n | n | y | n | y | n | n |
| Cohort Study of Young Girls' Nutrition, Environment, and Transitions (CYGNET) | Pre-pregnancy BMI (Kubo 2014) | y | y | u | y | n | n | y | n | n | n | n |
| Danish National Birth Cohort | Maternal diet in pregnancy (Maslova 2017; Maslova 2018; Maslova 2019; Zhu 2017a; Zhu 2017b); GDM severity; Pre-pregnancy BMI; (Zhu 2016) | y | y | y | y | y | y | y | n | n | n | y |
| Exploring Perinatal outcomes Among CHildren (EPOCH) study | Breastfeeding (Crume 2012) | y | y | y | y | y | u | y | n | n | n | u |
| First Affiliated Hospital of Kunming Medical University (2003-2011) | Pre-pregnancy BMI; maternal gestational weight gain; large for gestational age; offspring sex (Zhao 2015) | y | y | y | y | y | n | n | y | n | n | n |
| First Affiliated Hospital of Kunming Medical University (2003-2011) | Breastfeeding (Zhao 2015) | y | y | u | y | y | u | n | y | n | n | n |
| German GDM Offspring Study (1989-2000) | Pre-pregnancy BMI; offspring birthweight (Boerschmann 2010) | y | y | y | y | n | n | y | n | n | n | n |
| German GDM Offspring Study (1989-2000) | Smoking in pregnancy (Boerschmann 2010) | y | y | u | y | n | n | y | y | n | n | n |
| Growing up today in Singapore (GUSTO) study | Breastfeeding (Aris 2017) | y | y | y | y | n | u | y | y | n | n | n |
| Growing Up Today Study and Nurses' Health Study II | Offspring birthweight; offspring sex (Gillman 2003) | y | y | y | y | y | u | y | n | n | n | y |
| Institute of Nutrition and Functional Foods, Laval University | Breastfeeding (Dugas 2018) | y | y | y | y | n | n | y | n | n | n | n |
| Medical Birth Registry of Xiamen | Offspring birthweight (Chen 2020) | y | y | y | y | y | y | y | n | y | y | y |
| Medical Birth Registry of Xiamen | GDM severity (Shi 2020) | y | y | y | y | n | n | y | n | n | y | n |
| Mercy Hospital for Women, Melbourne (1971-2003) | Postpartum leptin, adiponectin (Shub 2019) | y | y | y | y | y | y | y | n | n | n | n |
| National Maternity Collection (MAT) | GDM treatment (Landi 2019) | y | y | y | y | y | y | y | n | n | n | n |
| Northern Finland Birth Cohort of 1986 | Pre-pregnancy BMI (Pirkola 2010) | y | y | y | y | n | n | y | n | n | n | n |
| PREOBE study | Cord blood metabolites (Shorky 2019) | y | y | y | y | n | n | y | y | n | n | n |
| Royal Victoria Hospital (1989-1991) | GDM severity (Egeland 2010) | y | y | y | y | n | n | y | n | n | n | n |
| SIPS | Preeclampsia (Huang 2020) | y | y | y | y | y | n | y | n | y | y | n |
| Study of Latino Adolescents at Risk for Diabetes (SOLAR) | Breastfeeding (Vandyousefi 2019) | y | y | n | y | y | n | y | n | n | n | n |
| SWIFT Offspring Study | Breastfeeding (Faith 2019, Gunderson 2018) | y | y | y | y | y | y | u | n | y | n | n |
| Tianjin Gestational Diabetes Mellitus Prevention Program | Pre-pregnancy BMI; gestational weight gain; (Leng 2015); Maternal obesity and glucose genetic risk scores (Liang 2020; Song 2021); GDM severity (Zhang 2015); Pregnancy hypertension (Zhang 2017) | y | y | y | y | y | y | y | n | n | n | n |
| University Medical Center Utrecht (2000-2006) | Birthweight; fetal anthropometrics (Hammoud 2017) | y | y | y | y | y | n | n | y | n | n | n |
| Vivantes Medical Center, Berlin | Pre-pregnancy BMI; GDM severity; Neonatal anthropometrics (Schaefer-Graf 2005) | y | y | y | y | n | n | y | n | y | n | n |
| Vivantes Medical Center, Berlin | Breastfeeding (Schaefer-Graf 2006) | y | y | y | y | n | n | y | n | y | n | n |
| Women and Infants Hospital, Rhode Island | Large for gestational age; pre-pregnancy BMI; maternal gestational weight gain; GDM severity (Vohr 1999) | y | y | y | y | y | n | y | n | n | n | n |
| Women and Infants Hospital, Rhode Island (1991-1993) | Large for gestational age; pre-pregnancy BMI; maternal gestational weight gain; GDM severity (Vohr 1997) | y | y | y | y | y | n | y | n | n | n | n |
| Women, Infants and Children Program in Los Angeles County | Breastfeeding (Shearrer 2015) | y | y | n | y | n | n | y | n | n | n | n |

**Fig. 4 Heat map of study quality according to the Diabetes Canada Clinical Practice Guidelines for observational studies assessing offspring anthropometric and cardiometabolic outcomes.** Green cells indicate high quality; red cells indicate low quality. Yellow cells indicate unclear/unable to assess quality based on information provided.

counterparts[5], studies among women with a history of GDM indicated dose-response associations of maternal BMI – primarily, pre-pregnancy BMI—and GDM severity with these endpoints. However, the extent to which these physiological factors are modifiable remains yet to be determined. Given the paucity of available research on CVD risk in women with a history of GDM, and the low certainty of evidence assessment, this is a research area ripe for investigation.

**Quality of maternal studies**. We ranked the quality of evidence for prognostic factors indicative of risk of T2D or CVD in women as Level 4 (low)[12]. Most empirical literature comes predominantly from large health care registries that boast large sample sizes and decades of follow-up. However, they carry high risk of bias in terms of identifying and interpretation specific prognostic characteristics as causal risk factors due to residual confounding due to maternal lifestyle, pre-existing chronic conditions, and other characteristics around time of pregnancy and GDM diagnoses. For example, although maternal hypertension during pregnancy may be a risk factor for T2D or CVD, the association may be explained by maternal BMI, diet quality, physical activity, smoking status, socioeconomic factors, and more. In contrast, there are notable large prospective cohorts, including CARDIA (e.g. refs. [31,46]) and the Nurses' Health Study II (e.g. refs. [47,67]), that collected detailed prospective information on the above-mentioned variables, thereby mitigating risk of bias in these studies.

**Offspring outcomes**. The most common measure of offspring anthropometry was BMI between 2 and 10 years after birth. As with maternal outcomes, observational evidence for offspring indicates that greater GDM severity and higher maternal pre-pregnancy BMI predicts higher offspring adiposity. Yet, interpretation of these findings should be tempered with results of intervention studies showing that GDM treatment did not affect offspring anthropometrics[77–79]. Other frequently studied

perinatal predictors of offspring adiposity included birth size and breastfeeding duration/exclusivity. Generally, higher birthweight tended to be associated with higher future BMI[55–57,65,66]. Some observational studies showed a protective effect of breastfeeding against offspring obesity risk during childhood, though this finding was not consistently observed. A few observational studies reported a modifying effect of offspring biological sex on future body composition among children exposed to GDM (e.g.[57,88]), but the direction of association was not consistent. Of the five RCTs that investigated the effect of GDM treatment on offspring anthropometry and body composition, two found that treatment with Metformin, as compared to insulin, predicted higher offspring adiposity according to skinfold thicknesses[80] and weight[81] within the first 18 months of life. These results call for additional research to assess long-term offspring outcomes related to pharmaceutical treatments for GDM, especially given findings indicating comparable neonatal outcomes among women treated with Metformin and insulin[89].

Most studies that assessed offspring cardiometabolic profile were observational and focused on prognostic factors that occurred during the perinatal/postpartum period, though a few RCTs targeting maternal glycemic control during pregnancy via pharmaceutical treatments and/or lifestyle alterations. Among observational studies (Supplementary Data 9), common prognostic factors included maternal BMI and diet, for which both prognostic factors yielded inconsistent associations with offspring cardiometabolic profile. As with the studies assessing offspring anthropometry and body composition as outcomes, RCTs to prevent GDM among high-risk women generally found minimal effects of the pharmaceutical and/or lifestyle interventions on offspring cardiometabolic profile (Supplementary Data 10). This, again, suggests that additional research is needed to better understand the pathophysiology of maternal GDM, to characterize relevant in utero programming pathways[90–92], and identify accurate and valid prognostic biomarkers—including those in

cord blood—as well as outcomes in offspring that are more relevant to future disease risk[6] such as directly-assessed neonatal adiposity[92].

**Quality of offspring studies**. As with the maternal studies, we categorized the literature on prognostic factors for offspring outcomes as being of low quality (Level 4)[12]. The inconsistent observational findings in conjunction with null results of RCTs targeting prevention of GDM among high-risk women indicate the existence of residual confounding for observational studies, and in the cases of the trials, the possibility that the interventions were developed with a suboptimal endpoint (e.g., a focus on preventing macrosomia based on birth size rather than directly assessed neonatal adiposity). Future work is needed to gain a better understanding of in utero programming mechanisms that may link maternal GDM to offspring adiposity, as well as interventions specifically formulated to prevent neonatal adiposity assessed via gold standard methods such as computed tomography or dual X-ray absorptiometry[93,94].

**Strengths and limitations of studies included in the systematic review**. A key strength of many studies included in this systematic review is the prospective study design, which enhances temporal and causal inference regarding prognostic capacity of the maternal and offspring characteristics and behaviors assessed in studies herein. Additional strengths of some, but not all studies, include multi-ethnic study populations, which enhance generalizability of findings; large sample sizes, which improves capacity to detect biologically relevant associations; and use of gold standard assessments of the maternal and offspring outcomes of interest.

Limitations include the low-grade quality of studies included in this review (residual confounding, reverse causation bias, attrition and selection bias, inadequate duration of follow-up). Additionally, most studies were not designed to explore the long-term prognosis of GDM. Accordingly, many studies comprise post hoc analyses that were likely underpowered to detect smaller but biologically relevant effects of prognostic risk factors solely among mothers and/or offspring exposed to GDM. When screening studies, we also noted that a general limitation of the literature on GDM prognostics in relation to offspring outcomes is assessment of the prognostic variable(s) of interest contemporaneously with outcome assessment, which limits our ability to make causal inference on the effect of the prognostic variable on outcomes of interest. These shortcomings resulted in high risk of bias and low quality of studies.

**Strengths and limitations of systematic review approach and methodology**. Strengths of the methodology for this systematic review include implementation of at least two independent reviews across all phases of the extraction and assessment process, with an additional review by a third independent reviewer to resolve conflicts; and adherence to well-established assessments of research quality and assessments of bias. Limitations include the exclusively qualitative synthesis of results—a necessity given the relatively small number of studies identified; and as with all systematic reviews, the potential for our conclusions to be impacted by publication bias.

**Future directions**. Given the low quality of evidence identified in this systematic review, there is need for prospective cohort studies in diverse populations with granular data collection on prognostic risk factors as well as clinical and subclinical outcomes. Additionally, high fidelity of follow-up across the lifecourse,

particularly during sensitive windows of development during which there is greater developmental plasticity to respond to external cues[95], will shed light on avenues for primordial and/or primary prevention. Finally, consideration of appropriate adjustment covariates depending on the specific prognostic risk factor of interest (e.g., there is discourse regarding whether maternal pre-pregnancy BMI should be included as a covariate in models where GDM severity is the prognostic factor of interest given that these variables share overlapping in utero programming pathways[6,91]); and appropriate causal inference and analytical approaches to address structural biases that afflict observational study designs[95,96].

As interest in the application of precision prognostics to improve health for women and offspring affected by GDM pregnancies grows, there remains a crucial need to establish foundational knowledge regarding traditional prognostic factors which, in turn, will enhance our ability to identify new prognostic biomarkers that improve risk stratification for unfavorable health outcomes among both women and children affected by GDM.

## Data availability

The data that support the findings of this study are derived from published, peer-reviewed manuscripts. The search terms used to retrieve studies are found in the Supplementary Data 1 and the list of included studies is described in Supplementary Data 2 and 3. The source data underlying Figs. 2–4 is provided in Supplementary Data 4 to 10. All other relevant data are available from the authors upon request.

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

## Acknowledgements

The ADA/EASD Precision Diabetes Medicine Initiative, within which this work was conducted, has received the following support: The Covidence license was funded by Lund University (Sweden) for which technical support was provided by Maria Björklund and Krister Aronsson (Faculty of Medicine Library, Lund University, Sweden). Administrative support was provided by Lund University (Malmö, Sweden), University of Chicago (IL, USA), and the American Diabetes Association (Washington D.C., USA). The Novo Nordisk Foundation (Hellerup, Denmark) provided grant support for in-person writing group meetings (PI: L Phillipson, University of Chicago, IL). We also thank Marie-France Hivert for her valuable insights and feedback on this work. In addition, the following individuals were funded by the following sources:

- *Zhila Semnani-Azad* is funded by Canadian Institutes of Health Research (CIHR) Fellowship.
- *Romy Gaillard* is funded by the Dutch Diabetes Foundation (grant number 2017.81.002), the Netherlands Organization for Health Research and Development (NWO, ZonMw, grant number 543003109; NWO, ZonMw VIDI 09150172110034) and from the European Union's Horizon 2020 research and innovation programme under the ERA-NET Cofund action (no 727565), EndObesity, ZonMW the Netherlands (no. 529051026).
- *Alice Hughes* is funded by a Wellcome Trust GW4-Clinical Academic PhD Fellowship [WT203918].
- *Kristen Boyle* is funded by R01DK117168.
- *Deirdre K. Tobias* is funded by ADA-1-19-JDF-115.
- *Wei Perng* is funded by American Diabetes Association (ADA)−7-22-ICTSPM-08 and National Institutes of Health (NIH) U01 DK134981.

Funders did not play any role in the design of this study or the interpretation of the findings.

## Author contributions

Z.S.A., R.G., A.E.H., K.E.B., D.K.T. and W.P. completed the review, extraction, and quality assessment of papers. Z.S.A., R.G. and W.P. drafted the initial version of the manuscript. A.E.H., K.E.B. and D.K.T. provided critical intellectual feedback. All authors approved the final version for publication.

## Competing interests

The authors declare no competing interests.

## Additional information

## ADA/EASD PMDI

Deirdre K. Tobias[1,7], Jordi Merino[8,9,10], Abrar Ahmad[11], Catherine Aiken[12,13], Jamie L. Benham[14], Dhanasekaran Bodhini[15], Amy L. Clark[16], Kevin Colclough[17], Rosa Corcoy[18,19,20], Sara J. Cromer[9,21,22], Daisy Duan[23], Jamie L. Felton[24,25,26], Ellen C. Francis[27], Pieter Gillard[28], Véronique Gingras[29,30], Romy Gaillard[31], Eram Haider[32], Alice Hughes[17], Jennifer M. Ikle[33,34], Laura M. Jacobsen[35], Anna R. Kahkoska[36], Jarno L. T. Kettunen[37,38,39], Raymond J. Kreienkamp[9,10,21,40], Lee-Ling Lim[41,42,43], Jonna M. E. Männistö[44,45], Robert Massey[32], Niamh-Maire Mclennan[46], Rachel G. Miller[47], Mario Luca Morieri[48,49], Jasper Most[50], Rochelle N. Naylor[51], Bige Ozkan[52,53], Kashyap Amratlal Patel[17], Scott J. Pilla[54,55], Katsiaryna Prystupa[56,57], Sridharan Raghavan[58,59], Mary R. Rooney[52,60], Martin Schön[56,57,61], Zhila Semnani-Azad[1✉], Magdalena Sevilla-Gonzalez[21,22,62], Pernille Svalastoga[63,64], Wubet Worku Takele[65], Claudia Ha-ting Tam[43,66,67], Anne Cathrine B. Thuesen[8], Mustafa Tosur[68,69,70], Amelia S. Wallace[52,60], Caroline C. Wang[60], Jessie J. Wong[71], Jennifer M. Yamamoto[72], Katherine Young[17], Chloé Amouyal[73,74], Mette K. Andersen[8], Maxine P. Bonham[75], Mingling Chen[76], Feifei Cheng[77], Tinashe Chikowore[22,78,79,80], Sian C. Chivers[81], Christoffer Clemmensen[8], Dana Dabelea[82], Adem Y. Dawed[32], Aaron J. Deutsch[10,21,22], Laura T. Dickens[83], Linda A. DiMeglio[24,25,26,84], Monika Dudenhöffer-Pfeifer[11], Carmella Evans-Molina[24,25,26,85], María Mercè Fernández-Balsells[86,87], Hugo Fitipaldi[11], Stephanie L. Fitzpatrick[88], Stephen E. Gitelman[89], Mark O. Goodarzi[90,91], Jessica A. Grieger[92,93], Marta Guasch-Ferré[1,94], Nahal Habibi[92,93], Torben Hansen[8], Chuiguo Huang[43,66], Arianna Harris-Kawano[24,25,26], Heba M. Ismail[24,25,26], Benjamin Hoag[95,96], Randi K. Johnson[97,98], Angus G. Jones[17,99], Robert W. Koivula[100], Aaron Leong[9,22,101], Gloria K. W. Leung[75], Ingrid M. Libman[102], Kai Liu[92], S. Alice Long[103], William L. Lowe Jr.[104], Robert W. Morton[105,106,107], Ayesha A. Motala[108], Suna Onengut-Gumuscu[109], James S. Pankow[110], Maleesa Pathirana[92,93], Sofia Pazmino[111], Dianna Perez[24,25,26], John R. Petrie[112], Camille E. Powe[9,21,22,113], Alejandra Quinteros[92], Rashmi Jain[114,115], Debashree Ray[60,116], Mathias Ried-Larsen[117,118], Zeb Saeed[119], Vanessa Santhakumar[7], Sarah Kanbour[54,120], Sudipa Sarkar[54], Gabriela S. F. Monaco[24,25,26], Denise M. Scholtens[121], Elizabeth Selvin[52,60], Wayne Huey-Herng Sheu[122,123,124], Cate Speake[125], Maggie A. Stanislawski[97], Nele Steenackers[111], Andrea K. Steck[126], Norbert Stefan[57,127,128], Julie Støy[129], Rachael Taylor[130], Sok Cin Tye[131,132], Gebresilasea Gendisha Ukke[65], Marzhan Urazbayeva[69,133], Bart Van der Schueren[111,134], Camille Vatier[135,136], John M. Wentworth[137,138,139], Wesley Hannah[140,141], Sara L. White[81,142], Gechang Yu[43,66], Yingchai Zhang[43,66], Shao J. Zhou[93,143], Jacques Beltrand[144,145], Michel Polak[144,145], Ingvild Aukrust[63,146], Elisa de Franco[17], Sarah E. Flanagan[17], Kristin A. Maloney[147], Andrew McGovern[17], Janne Molnes[63,146], Mariam Nakabuye[8], Pål Rasmus Njølstad[63,64], Hugo Pomares-Millan[11,148], Michele Provenzano[149], Cécile Saint-Martin[150], Cuilin Zhang[151,152], Yeyi Zhu[153,154], Sungyoung Auh[155], Russell de Souza[106,156], Andrea J. Fawcett[157,158], Chandra Gruber[159], Eskedar Getie Mekonnen[160,161], Emily Mixter[162], Diana Sherifali[106,163], Robert H. Eckel[164], John J. Nolan[165,166],

Louis H. Philipson[162], Rebecca J. Brown[155], Liana K. Billings[167,168], Kristen Boyle[82], Tina Costacou[47], John M. Dennis[17], Jose C. Florez[9,10,21,22], Anna L. Gloyn[33,34,169], Maria F. Gomez[11,170], Peter A. Gottlieb[126], Siri Atma W. Greeley[171], Kurt Griffin[115,172], Andrew T. Hattersley[17,99], Irl B. Hirsch[173], Marie-France Hivert[9,174,175], Korey K. Hood[71], Jami L. Josefson[157], Soo Heon Kwak[176], Lori M. Laffel[177], Siew S. Lim[65], Ruth J. F. Loos[8,178], Ronald C. W. Ma[43,66,67], Chantal Mathieu[28], Nestoras Mathioudakis[54], James B. Meigs[22,101,179], Shivani Misra[180,181], Viswanathan Mohan[182], Rinki Murphy[183,184,185], Richard Oram[17,99], Katharine R. Owen[100,186], Susan E. Ozanne[187], Ewan R. Pearson[32], Wei Perng[82], Toni I. Pollin[147,188], Rodica Pop-Busui[189], Richard E. Pratley[190], Leanne M. Redman[191], Maria J. Redondo[68,69], Rebecca M. Reynolds[46], Robert K. Semple[46,192], Jennifer L. Sherr[193], Emily K. Sims[24,25,26], Arianne Sweeting[194,195], Tiinamaija Tuomi[37,39,138], Miriam S. Udler[9,10,21,22], Kimberly K. Vesco[196], Tina Vilsbøll[197,198], Robert Wagner[56,57,199], Stephen S. Rich[109] & Paul W. Franks[1,11,100,107]

[7]Division of Preventative Medicine, Department of Medicine, Brigham and Women's Hospital and Harvard Medical School, Boston, MA, USA. [8]Novo Nordisk Foundation Center for Basic Metabolic Research, Faculty of Health and Medical Sciences, University of Copenhagen, Copenhagen, Denmark. [9]Diabetes Unit, Endocrine Division, Massachusetts General Hospital, Boston, MA, USA. [10]Center for Genomic Medicine, Massachusetts General Hospital, Boston, MA, USA. [11]Department of Clinical Sciences, Lund University Diabetes Centre, Lund University, Malmö, Sweden. [12]Department of Obstetrics and Gynaecology, the Rosie Hospital, Cambridge, UK. [13]NIHR Cambridge Biomedical Research Centre, University of Cambridge, Cambridge, UK. [14]Departments of Medicine and Community Health Sciences, Cumming School of Medicine, University of Calgary, Calgary, AB, Canada. [15]Department of Molecular Genetics, Madras Diabetes Research Foundation, Chennai, India. [16]Division of Pediatric Endocrinology, Department of Pediatrics, Saint Louis University School of Medicine, SSM Health Cardinal Glennon Children's Hospital, St. Louis, MO, USA. [17]Department of Clinical and Biomedical Sciences, University of Exeter Medical School, Exeter, Devon, UK. [18]CIBER-BBN, ISCIII, Madrid, Spain. [19]Institut d'Investigació Biomèdica Sant Pau (IIB SANT PAU), Barcelona, Spain. [20]Departament de Medicina, Universitat Autònoma de Barcelona, Bellaterra, Spain. [21]Programs in Metabolism and Medical & Population Genetics, Broad Institute, Cambridge, MA, USA. [22]Department of Medicine, Harvard Medical School, Boston, MA, USA. [23]Division of Endocrinology, Diabetes and Metabolism, Johns Hopkins University School of Medicine, Baltimore, MD, USA. [24]Department of Pediatrics, Indiana University School of Medicine, Indianapolis, IN, USA. [25]Herman B Wells Center for Pediatric Research, Indiana University School of Medicine, Indianapolis, IN, USA. [26]Center for Diabetes and Metabolic Diseases, Indiana University School of Medicine, Indianapolis, IN, USA. [27]Department of Biostatistics and Epidemiology, Rutgers School of Public Health, Piscataway, NJ, USA. [28]University Hospital Leuven, Leuven, Belgium. [29]Department of Nutrition, Université de Montréal, Montreal, Quebec, Canada. [30]Research Center, Sainte-Justine University Hospital Center, Montreal, Quebec, Canada. [31]Department of Pediatrics, Erasmus Medical Center, Rotterdam, The Netherlands. [32]Division of Population Health & Genomics, School of Medicine, University of Dundee, Dundee, UK. [33]Department of Pediatrics, Stanford School of Medicine, Stanford University, Stanford, CA, USA. [34]Stanford Diabetes Research Center, Stanford School of Medicine, Stanford University, Stanford, CA, USA. [35]University of Florida, Gainesville, FL, USA. [36]Department of Nutrition, University of North Carolina at Chapel Hill, Chapel Hill, NC, USA. [37]Helsinki University Hospital, Abdominal Centre/Endocrinology, Helsinki, Finland. [38]Folkhalsan Research Center, Helsinki, Finland. [39]Institute for Molecular Medicine Finland FIMM, University of Helsinki, Helsinki, Finland. [40]Department of Pediatrics, Division of Endocrinology, Boston Children's Hospital, Boston, MA, USA. [41]Department of Medicine, Faculty of Medicine, University of Malaya, Kuala Lumpur, Malaysia. [42]Asia Diabetes Foundation, Hong Kong SAR, China. [43]Department of Medicine & Therapeutics, Chinese University of Hong Kong, Hong Kong SAR, China. [44]Departments of Pediatrics and Clinical Genetics, Kuopio University Hospital, Kuopio, Finland. [45]Department of Medicine, University of Eastern Finland, Kuopio, Finland. [46]Centre for Cardiovascular Science, Queen's Medical Research Institute, University of Edinburgh, Edinburgh, UK. [47]Department of Epidemiology, University of Pittsburgh, Pittsburgh, PA, USA. [48]Metabolic Disease Unit, University Hospital of Padova, Padova, Italy. [49]Department of Medicine, University of Padova, Padova, Italy. [50]Department of Orthopedics, Zuyderland Medical Center, Sittard-Geleen, The Netherlands. [51]Departments of Pediatrics and Medicine, University of Chicago, Chicago, IL, USA. [52]Welch Center for Prevention, Epidemiology, and Clinical Research, Johns Hopkins Bloomberg School of Public Health, Baltimore, MD, USA. [53]Ciccarone Center for the Prevention of Cardiovascular Disease, Johns Hopkins School of Medicine, Baltimore, MD, USA. [54]Department of Medicine, Johns Hopkins University, Baltimore, MD, USA. [55]Department of Health Policy and Management, Johns Hopkins University Bloomberg School of Public Health, Baltimore, MD, USA. [56]Institute for Clinical Diabetology, German Diabetes Center, Leibniz Center for Diabetes Research at Heinrich Heine University Düsseldorf, Auf'm Hennekamp 65, 40225 Düsseldorf, Germany. [57]German Center for Diabetes Research (DZD), Ingolstädter Landstraße 1, 85764 Neuherberg, Germany. [58]Section of Academic Primary Care, US Department of Veterans Affairs Eastern Colorado Health Care System, Aurora, CO, USA. [59]Department of Medicine, University of Colorado School of Medicine, Aurora, CO, USA. [60]Department of Epidemiology, Johns Hopkins Bloomberg School of Public Health, Baltimore, MD, USA. [61]Institute of Experimental Endocrinology, Biomedical Research Center, Slovak Academy of Sciences, Bratislava, Slovakia. [62]Clinical and Translational Epidemiology Unit, Massachusetts General Hospital, Boston, MA, USA. [63]Mohn Center for Diabetes Precision Medicine, Department of Clinical Science, University of Bergen, Bergen, Norway. [64]Children and Youth Clinic, Haukeland University Hospital, Bergen, Norway. [65]Eastern Health Clinical School, Monash University, Melbourne, VIC, Australia. [66]Laboratory for Molecular Epidemiology in Diabetes, Li Ka Shing Institute of Health Sciences, The Chinese University of Hong Kong, Hong Kong, China. [67]Hong Kong Institute of Diabetes and Obesity, The Chinese University of Hong Kong, Hong Kong, China. [68]Department of Pediatrics, Baylor College of Medicine, Houston, TX, USA. [69]Division of Pediatric Diabetes and Endocrinology, Texas Children's Hospital, Houston, TX, USA. [70]Children's Nutrition Research Center, USDA/ARS, Houston, TX, USA. [71]Stanford University School of Medicine, Stanford, CA, USA. [72]Internal Medicine, University of Manitoba, Winnipeg, MB, Canada. [73]Department of Diabetology, APHP, Paris, France. [74]Sorbonne Université, INSERM, NutriOmic team, Paris, France. [75]Department of Nutrition, Dietetics and Food, Monash University, Melbourne, VIC, Australia. [76]Monash Centre for Health Research and Implementation, Monash University, Clayton, VIC, Australia. [77]Health Management Center, The Second Affiliated Hospital of Chongqing Medical University, Chongqing Medical University, Chongqing, China. [78]MRC/Wits Developmental Pathways for Health Research Unit, Department of Paediatrics, Faculty of Health Sciences, University of the Witwatersrand, Johannesburg, South Africa. [79]Channing Division of Network Medicine, Brigham and Women's Hospital, Boston, MA, USA. [80]Sydney Brenner Institute for Molecular Bioscience, Faculty of Health

Sciences, University of the Witwatersrand, Johannesburg, South Africa. [81]Department of Women and Children's health, King's College London, London, UK. [82]Lifecourse Epidemiology of Adiposity and Diabetes (LEAD) Center, University of Colorado Anschutz Medical Campus, Aurora, CO, USA. [83]Section of Adult and Pediatric Endocrinology, Diabetes and Metabolism, Kovler Diabetes Center, University of Chicago, Chicago, USA. [84]Department of Pediatrics, Riley Hospital for Children, Indiana University School of Medicine, Indianapolis, IN, USA. [85]Richard L. Roudebush VAMC, Indianapolis, IN, USA. [86]Biomedical Research Institute Girona, IdIBGi, Girona, Spain. [87]Diabetes, Endocrinology and Nutrition Unit Girona, University Hospital Dr Josep Trueta, Girona, Spain. [88]Institute of Health System Science, Feinstein Institutes for Medical Research, Northwell Health, Manhasset, NY, USA. [89]University of California at San Francisco, Department of Pediatrics, Diabetes Center, San Francisco, CA, USA. [90]Division of Endocrinology, Diabetes and Metabolism, Cedars-Sinai Medical Center, Los Angeles, CA, USA. [91]Department of Medicine, Cedars-Sinai Medical Center, Los Angeles, CA, USA. [92]Adelaide Medical School, Faculty of Health and Medical Sciences, The University of Adelaide, Adelaide, SA, Australia. [93]Robinson Research Institute, The University of Adelaide, Adelaide, SA, Australia. [94]Department of Public Health and Novo Nordisk Foundation Center for Basic Metabolic Research, Faculty of Health and Medical Sciences, University of Copenhagen, 1014 Copenhagen, Denmark. [95]Division of Endocrinology and Diabetes, Department of Pediatrics, Sanford Children's Hospital, Sioux Falls, SD, USA. [96]University of South Dakota School of Medicine, E Clark St, Vermillion, SD, USA. [97]Department of Biomedical Informatics, University of Colorado Anschutz Medical Campus, Aurora, CO, USA. [98]Department of Epidemiology, Colorado School of Public Health, Aurora, CO, USA. [99]Royal Devon University Healthcare NHS Foundation Trust, Exeter, UK. [100]Oxford Centre for Diabetes, Endocrinology and Metabolism, University of Oxford, Oxford, UK. [101]Division of General Internal Medicine, Massachusetts General Hospital, Boston, MA, USA. [102]UPMC Children's Hospital of Pittsburgh, Pittsburgh, PA, USA. [103]Center for Translational Immunology, Benaroya Research Institute, Seattle, WA, USA. [104]Department of Medicine, Northwestern University Feinberg School of Medicine, Chicago, IL, USA. [105]Department of Pathology & Molecular Medicine, McMaster University, Hamilton, ON, Canada. [106]Population Health Research Institute, Hamilton, ON, Canada. [107]Department of Translational Medicine, Medical Science, Novo Nordisk Foundation, Tuborg Havnevej 19, 2900 Hellerup, Denmark. [108]Department of Diabetes and Endocrinology, Nelson R Mandela School of Medicine, University of KwaZulu-Natal, Durban, South Africa. [109]Center for Public Health Genomics, Department of Public Health Sciences, University of Virginia, Charlottesville, VA, USA. [110]Division of Epidemiology and Community Health, School of Public Health, University of Minnesota, Minneapolis, MN, USA. [111]Department of Chronic Diseases and Metabolism, Clinical and Experimental Endocrinology, KU Leuven, Leuven, Belgium. [112]School of Health and Wellbeing, College of Medical, Veterinary and Life Sciences, University of Glasgow, Glasgow, UK. [113]Department of Obstetrics, Gynecology, and Reproductive Biology, Massachusetts General Hospital and Harvard Medical School, Boston, MA, USA. [114]Sanford Children's Specialty Clinic, Sioux Falls, SD, USA. [115]Department of Pediatrics, Sanford School of Medicine, University of South Dakota, Sioux Falls, SD, USA. [116]Department of Biostatistics, Johns Hopkins Bloomberg School of Public Health, Baltimore, MD, USA. [117]Centre for Physical Activity Research, Rigshospitalet, Copenhagen, Denmark. [118]Institute for Sports and Clinical Biomechanics, University of Southern Denmark, Odense, Denmark. [119]Department of Medicine, Division of Endocrinology, Diabetes and Metabolism, Indiana University School of Medicine, Indianapolis, IN, USA. [120]AMAN Hospital, Doha, Qatar. [121]Department of Preventive Medicine, Division of Biostatistics, Northwestern University Feinberg School of Medicine, Chicago, IL, USA. [122]Institute of Molecular and Genomic Medicine, National Health Research Institutes, Taipei City, Taiwan, ROC. [123]Divsion of Endocrinology and Metabolism, Taichung Veterans General Hospital, Taichung, Taiwan, ROC. [124]Division of Endocrinology and Metabolism, Taipei Veterans General Hospital, Taipei, Taiwan, ROC. [125]Center for Interventional Immunology, Benaroya Research Institute, Seattle, WA, USA. [126]Barbara Davis Center for Diabetes, University of Colorado Anschutz Medical Campus, Aurora, CO, USA. [127]University Hospital of Tübingen, Tübingen, Germany. [128]Institute of Diabetes Research and Metabolic Diseases (IDM), Helmholtz Center Munich, Neuherberg, Germany. [129]Steno Diabetes Center Aarhus, Aarhus University Hospital, Aarhus, Denmark. [130]University of Newcastle, Newcastle upon Tyne, UK. [131]Sections on Genetics and Epidemiology, Joslin Diabetes Center, Harvard Medical School, Boston, MA, USA. [132]Department of Clinical Pharmacy and Pharmacology, University Medical Center Groningen, Groningen, The Netherlands. [133]Gastroenterology, Baylor College of Medicine, Houston, TX, USA. [134]Department of Endocrinology, University Hospitals Leuven, Leuven, Belgium. [135]Sorbonne University, Inserm U938, Saint-Antoine Research Centre, Institute of Cardiometabolism and Nutrition, Paris 75012, France. [136]Department of Endocrinology, Diabetology and Reproductive Endocrinology, Assistance Publique-Hôpitaux de Paris, Saint-Antoine University Hospital, National Reference Center for Rare Diseases of Insulin Secretion and Insulin Sensitivity (PRISIS), Paris, France. [137]Department of Diabetes and Endocrinology, Royal Melbourne Hospital, Parkville, VIC, Australia. [138]Walter and Eliza Hall Institute, Parkville, VIC, Australia. [139]Department of Medicine, University of Melbourne, Parkville, VIC, Australia. [140]Deakin University, Melbourne, VIC, Australia. [141]Department of Epidemiology, Madras Diabetes Research Foundation, Chennai, India. [142]Department of Diabetes and Endocrinology, Guy's and St Thomas' Hospitals NHS Foundation Trust, London, UK. [143]School of Agriculture, Food and Wine, University of Adelaide, Adelaide, SA, Australia. [144]Institut Cochin, Inserm U 10116, Paris, France. [145]Pediatric endocrinology and diabetes, Hopital Necker Enfants Malades, APHP Centre, université de Paris, Paris, France. [146]Department of Medical Genetics, Haukeland University Hospital, Bergen, Norway. [147]Department of Medicine, University of Maryland School of Medicine, Baltimore, MD, USA. [148]Department of Epidemiology, Geisel School of Medicine at Dartmouth, Hanover, NH, USA. [149]Nephrology, Dialysis and Renal Transplant Unit, IRCCS—Azienda Ospedaliero-Universitaria di Bologna, Alma Mater Studiorum University of Bologna, Bologna, Italy. [150]Department of Medical Genetics, AP-HP Pitié-Salpêtrière Hospital, Sorbonne University, Paris, France. [151]Global Center for Asian Women's Health, Yong Loo Lin School of Medicine, National University of Singapore, Singapore, Singapore. [152]Department of Obstetrics and Gynecology, Yong Loo Lin School of Medicine, National University of Singapore, Singapore, Singapore. [153]Division of Research, Kaiser Permanente Northern California, Oakland, CA, USA. [154]Department of Epidemiology and Biostatistics, University of California San Francisco, San Francisco, CA, USA. [155]National Institute of Diabetes and Digestive and Kidney Diseases, National Institutes of Health, Bethesda, MD, USA. [156]Department of Health Research Methods, Evidence, and Impact, Faculty of Health Sciences, McMaster University, Hamilton, ON, Canada. [157]Ann & Robert H. Lurie Children's Hospital of Chicago, Department of Pediatrics, Northwestern University Feinberg School of Medicine, Chicago, IL, USA. [158]Department of Clinical and Organizational Development, Chicago, IL, USA. [159]American Diabetes Association, Arlington, VA, USA. [160]College of Medicine and Health Sciences, University of Gondar, Gondar, Ethiopia. [161]Global Health Institute, Faculty of Medicine and Health Sciences, University of Antwerp, 2160 Antwerp, Belgium. [162]Department of Medicine and Kovler Diabetes Center, University of Chicago, Chicago, IL, USA. [163]School of Nursing, Faculty of Health Sciences, McMaster University, Hamilton, ON, Canada. [164]Division of Endocrinology, Metabolism, Diabetes, University of Colorado, Boulder, CO, USA. [165]Department of Clinical Medicine, School of Medicine, Trinity College Dublin, Dublin, Ireland. [166]Department of Endocrinology, Wexford General Hospital, Wexford, Ireland. [167]Division of Endocrinology, NorthShore University HealthSystem, Skokie, IL, USA. [168]Department of Medicine, Prtizker School of Medicine, University of Chicago, Chicago, USA. [169]Department of Genetics, Stanford School of Medicine, Stanford University, Stanford, CA, USA. [170]Faculty of Health, Aarhus University, Aarhus, Denmark. [171]Departments of Pediatrics and Medicine and Kovler Diabetes Center, University of Chicago, Chicago, IL, USA. [172]Sanford Research, Sioux Falls, SD, USA. [173]University of Washington School of Medicine, Seattle, WA, USA. [174]Department of Population Medicine, Harvard Medical School, Harvard Pilgrim Health Care Institute, Boston, MA, USA. [175]Department of Medicine, Universite de Sherbrooke, Sherbrooke, QC, Canada. [176]Department of Internal Medicine, Seoul National University College of Medicine, Seoul National University Hospital, Seoul, Republic of Korea. [177]Joslin Diabetes

Center, Harvard Medical School, Boston, MA, USA. [178]Charles Bronfman Institute for Personalized Medicine, Icahn School of Medicine at Mount Sinai, New York, NY, USA. [179]Broad Institute, Cambridge, MA, USA. [180]Division of Metabolism, Digestion and Reproduction, Imperial College London, London, UK. [181]Department of Diabetes & Endocrinology, Imperial College Healthcare NHS Trust, London, UK. [182]Department of Diabetology, Madras Diabetes Research Foundation & Dr. Mohan's Diabetes Specialities Centre, Chennai, India. [183]Department of Medicine, Faculty of Medicine and Health Sciences, University of Auckland, Auckland, New Zealand. [184]Auckland Diabetes Centre, Te Whatu Ora Health New Zealand, Auckland, New Zealand. [185]Medical Bariatric Service, Te Whatu Ora Counties, Health New Zealand, Auckland, New Zealand. [186]Oxford NIHR Biomedical Research Centre, University of Oxford, Oxford, UK. [187]University of Cambridge, Metabolic Research Laboratories and MRC Metabolic Diseases Unit, Wellcome-MRC Institute of Metabolic Science, Cambridge, UK. [188]Department of Epidemiology & Public Health, University of Maryland School of Medicine, Baltimore, MD, USA. [189]Department of Internal Medicine, Division of Metabolism, Endocrinology and Diabetes, University of Michigan, Ann Arbor, MI, USA. [190]AdventHealth Translational Research Institute, Orlando, FL, USA. [191]Pennington Biomedical Research Center, Baton Rouge, LA, USA. [192]MRC Human Genetics Unit, Institute of Genetics and Cancer, University of Edinburgh, Edinburgh, UK. [193]Yale School of Medicine, New Haven, CT, USA. [194]Faculty of Medicine and Health, University of Sydney, Sydney, NSW, Australia. [195]Department of Endocrinology, Royal Prince Alfred Hospital, Sydney, NSW, Australia. [196]Kaiser Permanente Northwest, Kaiser Permanente Center for Health Research, Portland, OR, USA. [197]Clinial Research, Steno Diabetes Center Copenhagen, Herlev, Denmark. [198]Department of Clinical Medicine, Faculty of Health and Medical Sciences, University of Copenhagen, Copenhagen, Denmark. [199]Department of Endocrinology and Diabetology, University Hospital Düsseldorf, Heinrich Heine University Düsseldorf, Moorenstr. 5, 40225 Düsseldorf, Germany.

