## [Peer Review File · Communications Medicine]

Reviewers' comments:

Reviewer #1 (Remarks to the Author):

This paper is a systematic review that aim to synthetise evidence on prognostic conditions, risk factors and biomarkers among women and children affected by gestational diabetes (GDM) during pregnancies, focusing on the clinical endpoints of cardiovascular diseases (CVD) and type 2 diabetes (T2D) among women and adiposity and cardiometabolic risk profile among offspring exposed to GDM in utero.

This paper is very well designed and written, and follow the recommendations for such research. The search strategy systematic review protocol were published to PROSPERO (CRD42021276094). This Systematic Review is written in support of the 2nd International Consensus Report on Precision Diabetes Medicine.

The subject is very relevant as there are many publications about maternal and offspring outcomes after GDM during pregnancy and it is important to identify prognostic factors and biomarkers among women with a history of GDM and/or their children who were exposed to GDM in utero.

There are some minor issues that need to be clarify and the numbering of the references in the text needs to be revised for certain sections as they do not correspond to the tables.

Method:

This part is well described. Anyway, it is not clear if you have excluded adult offspring and why. Can you clarify this point?

Results:

Maternal outcomes

Maternal T2D:

P9 line 134 : the data on post-partum weight gain (ref 12) is reported in the table 2.

P9 line 139: the RCT cited (ref 17) is not in table 3

P10 lines 148-151 : the ref 36 (RCT) is not in table 3 and the number for ref 37-40 do not match with those reported in the table 2.

Line 149 : could you precise if dietary pattern was assessed during life course or during pregnancy.

Line 150: is it clinical or biological biomarkers? Can you provide details about the biomarkers and the period when they were explored?

Maternal cardiovascular diseases:

The references numbers in the text must be checked because some of them don't match with those in table 4 (for ex, ref 17 does not include maternal BMI)

Can you report results about BMI during life course in the text? (ref 43).

For diet and smoking, can you provide details on the period of life for the exposition (pregnancy or life course).

Offspring outcomes

Anthropometry:

P11 line 173: you can add that for the RCT, the association with type of treatment during pregnancy was also considered.

P11 line 181-183: the sentence must be revised.

Cardiometabolic profile:

P12 line 206: there is only one observational study that is not consistent.

P12 line 207: in table 8, there is only one RCT (ref 105) that compared diet vs insulin. But, in this study, the intervention was metformin versus insulin, not diet versus insulin.

Discussion

The paragraph 4.2.2 about maternal T2D should be before the paragraph 4.2.1, in the same order as the results.

P14 line 241: are at higher risk for CVD

P14 line 243: you can add the period of the exposition for maternal BMI (before, during pregnancy or during the life course?)

P16 paragraph 4.3.1: there is an issue about the age at follow-up for the offspring (see above). Did you search for adults? If yes and you did not find data about them, you could discuss this point. If not, you should justify this.

P17 lines 302-303: you could add that biomarkers should be studied in future studies.

P17 line 313: you could underline that you did not find any study about biomarkers in the offspring, particularly biomarkers at birth (cord blood) that could be predictive markers for risk of adverse outcomes.

P17 line 320: what do you mean by gold standard assessments?

A table synthesising the main exposures identified for each outcomes and the level of evidence will help the reader.

Table 1: Numbering of the references is right. But, there is 37 references in the table, not 35 references, for offspring outcomes.

Table 2:

Eades 2015 (p 43) : this paper also reported on postpartum GWG according to the text.

Tobias 2012 (p 44): can you give precision on the type of diet?

Schaefer-Graf 2009 (p 45): the novel biomarker is in fact a clinical composite score.

Table 4:

Pintaudi 2015: can you specify pregnancy complications. It seems it is stillbirth.

Table 6, 7 and 8: age of outcome assessment should be added.

Table 8:

Battin 2015: the intervention was metformin versus insulin, not diet versus insulin.

Figure 1: can you add in the flow chart the reasons why studies were excluded at the two first steps. You should also indicate at the end of the flow chart the number of studies included in the review for women and the number for offspring, separately.

Figure 2, 3 and 4: can you add the number of the references corresponding to each line.

Figure 3: I understood that the first line refers to ref 85 and 76 and then it also includes offspring outcomes.

Reviewer #2 (Remarks to the Author):

This compilation of the evidence on GDM and progression to type 2 diabetes, cardiometabolic outcomes and risk of cardiovascular disease provides an important contribution to the literature. The assembly of the publications is well-organized and mostly reasonable. However, the authors have omitted several studies that make important contributions to the literature, and have the highest quality of design. For example, the rigorous systematic repeated testing of cardiometabolic profiles in both SWIFT and CARDIA are rare. In addition, CARDIA contributes research measures of cardiovascular disease risk and full glucose tolerance many years after pregnancy. Several publications from these diverse prospective USA research cohorts, the SWIFT Study and the CARDIA Study have been omitted from the review.

The CARDIA Study published several studies on GDM and cardiovascular health outcomes, including how glucose tolerance may affect cIMT and CAC measures of CVD risk.

The SWIFT Study includes: SWIFT Women Study (n=1033), the SWIFT Offspring Study of infants (n=550), and the SWIFT Study in Youth (n=1033) children. A key study on early diet in SWIFT children was omitted and is attached below. In SWIFT Women, there are also several publications on metabolomics and lipidomics from SWIFT were also omitted from this review. The list of these studies and their features (summarized below), as well as the supplemental files and a manuscript are attached.

For a comprehensive evaluation of the studies, it would be important to state identify studies that obtained direct measures of "GDM severity" from the EHR, and that most studies of GDM offspring simply classify individuals as GDM "yes or no". Also, the CARDIA study measured glucose tolerance before pregnancies, and thus, is one of the only studies that could rule out diabetes before pregnancy. This feature should be emphasized in the review. Gunderson et al. AJE 2010 analyzed fasting cardiometabolic parameters that predicted GDM. The standardized preconception measures of risk factors is a unique strength of CARDIA.

A recent review paper shown below cites the relevant manuscripts published on GDM, T2D and CVD outcomes from CARDIA:

Kim C, Catov J, Schreiner PJ, Appiah D, Wellons MF, Siscovick D, Calderon-Margalit R, Huddleston H, Ebong IA, Lewis CE. Women's Reproductive Milestones and Cardiovascular Disease Risk: A Review of Reports and Opportunities From the CARDIA Study. *J Am Heart Assoc.* 2023 Mar 7;12(5):e028132. doi: 10.1161/JAHA.122.028132. Epub 2023 Feb 27. PMID: 36847077; PMCID: PMC10111436.

Details about the publications to include:

1. Findings from the CARDIA have been omitted from Table 2 that shows the prognostic factors associated with type 2 diabetes.

A. The following paper includes a table showing the independent associations between maternal BMI, lifestyle behaviors, race disparities and other factors related to type 2 diabetes progression -- lactation was the main effect, but we reported other factors that influence progression. I attach the documents with the Relative Hazards for the factors above. See Supplemental Figure 3 attached pdf.

Reference: Gunderson EP, Lewis CE, Lin Y, Sorel M, Gross M, Sidney S, Jacobs DR Jr, Shikany JM, Quesenberry CP Jr. Lactation Duration and Progression to Diabetes in Women Across the Childbearing Years: The 30-Year CARDIA Study. *JAMA Intern Med.* 2018 Mar 1;178(3):328-337. doi: 10.1001/jamainternmed.2017.7978. PMID: 29340577; PMCID: PMC5885916.

2. Several studies have been omitted from Table 4 on the prognostic factors for cardiovascular disease among women with GDM

A. This paper show below evaluates the impact of progression to normoglycemia, prediabetes and diabetes after GDM and non-GDM associated with risk of Coronary Artery Calcium, a strong risk factor for CVD in young adults.

Reference: Gunderson EP, Sun B, Catov JM, Carnethon M, Lewis CE, Allen NB, Sidney S, Wellons M, Rana JS, Hou L, Carr JJ. Gestational Diabetes History and Glucose Tolerance After Pregnancy Associated With Coronary Artery Calcium in Women During Midlife: The CARDIA Study. *Circulation*. 2021 Mar 9;143(10):974-987. doi: 10.1161/CIRCULATIONAHA.120.047320. Epub 2021 Feb 1. PMID: 33517667; PMCID: PMC7940578.

B. This paper evaluates the relationship between GDM and no GDM with and without prediabetes and CVD risk factors related to carotid artery IMT.

Reference: Gunderson EP, Chiang V, Pletcher MJ, Jacobs DR, Quesenberry CP, Sidney S, Lewis CE. History of gestational diabetes mellitus and future risk of atherosclerosis in mid-life: the Coronary Artery Risk Development in Young Adults study. *J Am Heart Assoc*. 2014 Mar 12;3(2):e000490. doi: 10.1161/JAHA.113.000490. PMID: 24622610; PMCID: PMC4187501.

C. This paper evaluates changes in cardiometabolic risk factors related to GDM

Reference: Catov JM, Sun B, Bertolet M, Snyder GG, Lewis CE, Allen NB, Shikany JM, Ingram KH, Appiah D, Gunderson EP. Changes in Cardiometabolic Risk Factors Before and After Gestational Diabetes: A Prospective Life-Course Analysis in CARDIA Women. *Obesity (Silver Spring)*. 2020 Aug;28(8):1397-1404. doi: 10.1002/oby.22848. Epub 2020 Jul 6. PMID: 32627948; PMCID: PMC7501158.

3. The review also omitted an important prospective, longitudinal study from the SWIFT Study in Children (n=844)

A. This longitudinal study from pregnancy, birth and up to age 2-5 years evaluated the impact of maternal risk factors, GDM severity and detailed early infant diet (sugar sweetened beverages, natural fruit juice) from birth to age 1 years and increase subsequent risk of child obesity at 2-5 years. This SWIFT study is a high diverse cohort, and includes with 25% low income families. It is much stronger than the WIC cross-sectional study of children that had diet and BMI assessed at a single time point. This is an important analysis, and one of the only studies published that evaluated infant diet and breastfeeding, specifically sugary beverages and fruit juice from birth to 1 year.

Reference: Vandyousefi S, Davis JN, Gunderson EP. Association of infant diet with subsequent obesity at 2-5 years among children exposed to gestational diabetes: the SWIFT study. *Diabetologia*. 2021 May;64(5):1121-1132. doi: 10.1007/s00125-020-05379-y. Epub 2021 Jan 26. PMID: 33495846; PMCID: PMC8016720 -- detailed prospectively monthly surveys to assess infant feeding -- milk and all dietary intake of foods and beverages. Follow up with EHR measurements of child BMI.

4. There are also several manuscript from the SWIFT Study on the prediction of type 2 diabetes in a prospective cohort study of 1010 women with GDM who were tested with 2 h 75 g OGTT from 6-9 weeks postpartum and annually tested for progression to type 2 diabetes. We have conducted several nested case control studies (largest # cases of incident diabetes) in a single prospective GDM cohort in follow up from early postpartum. Only one of these studies was cited. We list a few of the other key

studies that have been innovative for metabolomic and lipidomic studies of pathways from GDM to future type 2 diabetes. These are important studies.

A. Allalou A, Nalla A, Prentice KJ, Liu Y, Zhang M, Dai FF, Ning X, Osborne LR, Cox BJ, Gunderson EP, Wheeler MB. A Predictive Metabolic Signature for the Transition From Gestational Diabetes Mellitus to Type 2 Diabetes. *Diabetes*. 2016 Sep;65(9):2529-39. doi: 10.2337/db15-1720. Epub 2016 Jun 23. PMID: 27338739; PMCID: PMC5001181.

B. Lai M, Al Rijjal D, Röst HL, Dai FF, Gunderson EP, Wheeler MB. Underlying dyslipidemia postpartum in women with a recent GDM pregnancy who develop type 2 diabetes. *Elife*. 2020 Aug 4;9:e59153. doi: 10.7554/eLife.59153. PMID: 32748787; PMCID: PMC7417169.

C. Lai M, Liu Y, Ronnett GV, Wu A, Cox BJ, Dai FF, Röst HL, Gunderson EP, Wheeler MB. Amino acid and lipid metabolism in post-gestational diabetes and progression to type 2 diabetes: A metabolic profiling study. *PLoS Med*. 2020 May 20;17(5):e1003112. doi: 10.1371/journal.pmed.1003112. PMID: 32433647; PMCID: PMC7239388.

PUBLICATIONS FROM THE CARDIA AND SWIFT STUDIES THAT WERE NOT CITED AND ARE PROSPECTIVE STUDIES WITH SYSTEMATIC TESTING FOR TYPE 2 DIABETES, THE METABOLIC SYNDROME AND CARDIOVASCULAR DISEASE IN WOMEN WITH AND WITHOUT GDM, AND METABOLOMICS.

Recommend that Authors Cite these Publications: Original Research on Future Type 2 diabetes and Cardiovascular disease risk Associated with GDM in Women and their Children:

CARDIA PUBLICATIONS – Longitudinal, prospective cohorts studies on GDM status and type 2 diabetes, cardiometabolic health, and CVD risk. The are others on GDM and heart health.

1. CARDIA 2007– GDM AND NO GDM AND INCIDENT DIABETES AT 20-year followup.

Gunderson EP, Lewis CE, Tsai AL, Chiang V, Carnethon M, Quesenberry CP Jr, Sidney S. A 20-year prospective study of childbearing **and incidence of diabetes in young women, controlling for glycemia before conception:** the Coronary Artery Risk Development in Young Adults (CARDIA) Study. *Diabetes*. 2007 Dec;56(12):2990-6. doi: 10.2337/db07-1024. Epub 2007 Sep 26. PMID: 17898128; PMCID: PMC2952440. RH of incident diabetes according to GDM, no GDM, or no births. Incidence rates for GDM show 4% per year conversion to T2D in women during 20 year follow up. Only study to test for diabetes before conception to rule out T2D before GDM for accurate rates of progression.

CARDIA 2021– GDM AND NO GDM and subsequent risk of Coronary Artery Calcium by glucose tolerance (normal, prediabetes, or overt diabetes) during 25 year follow up

2. Gunderson EP, Sun B, Catov JM, Carnethon M, Lewis CE, Allen NB, Sidney S, Wellons M, Rana JS, Hou L, Carr JJ. **Gestational Diabetes History and Glucose Tolerance After Pregnancy Associated With Coronary Artery Calcium in Women During Midlife:** The CARDIA Study. *Circulation*. 2021 Mar 9;143(10):974-987. doi: 10.1161/CIRCULATIONAHA.120.047320. Epub 2021 Feb 1. PMID: 33517667; PMCID: PMC7940578. Relative

CARDIA 2007– GDM AND NO GDM and Subsequent Carotid Artery Intima Medial Thickness .

3. Gunderson EP, Chiang V, Pletcher MJ, Jacobs DR, Quesenberry CP, Sidney S, Lewis CE. **History of gestational diabetes mellitus and future risk of atherosclerosis in mid-life:** the Coronary Artery Risk Development in Young Adults study. *J Am Heart Assoc*. 2014 Mar 12;3(2):e000490. doi: 10.1161/JAHA.113.000490. PMID: 24622610; PMCID: PMC4187501.

CARDIA 2009– GDM AND NO GDM and Subsequent Incident Metabolic Syndrome

4. Gunderson EP, Jacobs DR Jr, Chiang V, Lewis CE, Tsai A, Quesenberry CP Jr, Sidney S. **Childbearing is associated with higher incidence of the metabolic syndrome among women of reproductive age controlling for measurements before pregnancy:** the CARDIA study. *Am J Obstet Gynecol*. 2009 Aug;201(2):177.e1-9. doi: 10.1016/j.ajog.2009.03.031. PMID: 19560114; PMCID: PMC2807822. Includes a comparison of Relative Hazards of incident MetS after GDM vs. No GDM pregnancies
5. Catov JM, Sun B, Bertolet M, Snyder GG, Lewis CE, Allen NB, Shikany JM, Ingram KH, Appiah D, Gunderson EP. **Changes in Cardiometabolic Risk Factors Before and After Gestational Diabetes:** A Prospective Life-Course Analysis in CARDIA Women. *Obesity (Silver Spring)*. 2020 Aug;28(8):1397-1404. doi: 10.1002/oby.22848. Epub 2020 Jul 6. PMID: 32627948; PMCID: PMC7501158.

CARDIA is unique among these pregnancy-related outcome studies in having preconception biochemical testing for diabetes in women about every 5 years from age 18-30 years. This study can rule out type 2 diabetes before pregnancy to identify specifically GDM. Other cohorts do not have systematic testing for diabetes among all women during the childbearing years. The NHS had relied on self-report of diabetes which would be considered to cover the majority after age 45 years.

RECOMMEND THAT THE AUTHORS CITE THE PRIMARY SOURCE OF RESEARCH ON EARLY LIFE DIET IN CHILDREN OF MOTHER WITH GDM FROM THE SWIFT STUDY IN YOUTH (N=1033 mother-child dyads born in 2008-2011) –this is the SWIFT entire cohort in follow up via the KP EHR. Much larger than the SWIFT Offspring Study during Infancy – detailed diet on all children. This is an ongoing longitudinal, prospective study in the entire SWIFT cohort of mothers and their children in follow up. Detailed prospective assessment of infant diet --- all components and milk feeding, and BMI measurements through 2023

SWIFT CHILDREN n =844 mother infant dyads followed during pregnancy and from birth up to 2 to 5 years

1. Vandyousefi S, Davis JN, Gunderson EP. **Association of infant diet with subsequent obesity at 2-5 years among children exposed to gestational diabetes: the SWIFT study.** Diabetologia. 2021 May;64(5):1121-1132. doi: 10.1007/s00125-020-05379-y. Epub 2021 Jan 26. PMID: 33495846; PMCID: PMC8016720. **SWIFT STUDY: Prospective longitudinal study of infant diet from birth and subsequent obesity at ages 2-5 years. Primary source of the research on breastfeeding, sugar sweetened beverages and fruit juice intake from birth through the first year of life – findings that very early exposure before 1 year of age is related to 3 to 10-fold higher future risk of child obesity. This is a prospective cohort study of SWIFT children (highly diverse cohort) followed from birth to age 2-5 years with monthly dietary assessments and measurement of size at birth and measured BMI at 2 to 5 years later.**

METABOLOMICS short- and long-term progression to Type 2 diabetes (SELECTED SWIFT PUBLICATIONS)

2. Allalou A, Nalla A, Prentice KJ, Liu Y, Zhang M, Dai FF, Ning X, Osborne LR, Cox BJ, Gunderson EP, Wheeler MB. **A Predictive Metabolic Signature for the Transition From Gestational Diabetes Mellitus to Type 2 Diabetes.** Diabetes. 2016 Sep;65(9):2529-39. doi: 10.2337/db15-1720. Epub 2016 Jun 23. PMID: 27338739; PMCID: PMC5001181.
3. Lai M, Al Rijjal D, Röst HL, Dai FF, Gunderson EP, Wheeler MB. **Underlying dyslipidemia postpartum in women with a recent GDM pregnancy who develop type 2 diabetes.** Elife. 2020 Aug 4;9:e59153. doi: 10.7554/eLife.59153. PMID: 32748787; PMCID: PMC7417169.
4. Lai M, Liu Y, Ronnett GV, Wu A, Cox BJ, Dai FF, Röst HL, Gunderson EP, Wheeler MB. **Amino acid and lipid metabolism in post-gestational diabetes and progression to type 2 diabetes: A metabolic profiling study.** PLoS Med. 2020 May 20;17(5):e1003112. doi: 10.1371/journal.pmed.1003112. PMID: 32433647; PMCID: PMC7239388.

**There are several more publications on metabolomics and lipidomics from the SWIFT Study
Senior Co-suthors Michael Wheeler and Erica P. Gunderson**

Reviewer #1

- Comment:** This paper is a systematic review that aim to synthetise evidence on prognostic conditions, risk factors and biomarkers among women and children affected by gestational diabetes (GDM) during pregnancies, focusing on the clinical endpoints of cardiovascular diseases (CVD) and type 2 diabetes (T2D) among women and adiposity and cardiometabolic risk profile among offspring exposed to GDM in utero. This paper is very well designed and written, and follow the recommendations for such research. The search strategy systematic review protocol were published to PROSPERO (CRD42021276094). This Systematic Review is written in support of the 2nd International Consensus Report on Precision Diabetes Medicine. The subject is very relevant as there are many publications about maternal and offspring outcomes after GDM during pregnancy and it is important to identify prognostic factors and biomarkers among women with a history of GDM and/or their children who were exposed to GDM in utero. There are some minor issues that need to be clarify and the numbering of the references in the text needs to be revised for certain sections as they do not correspond to the tables.

Response: We are grateful for the reviewer’s time and critical appraisal of this work. We have addressed comments/concerns and made appropriate changes throughout the manuscript (denoted by highlighted text) and tables, as described in the point-by-point response below.

Methods:

- Comment:** Method: This part is well described. Anyway, it is not clear if you have excluded adult offspring and why. Can you clarify this point?

Response: We did not place any exclusions of age of offspring and included adult offspring. To avoid confusion, we have replaced “children” with “offspring” throughout.

Results for Maternal T2D:

- Comment:** P9 line 134: the data on post-partum weight gain (ref 12) is reported in the Table 2.

Response: Correct; this study (Eades 2015) is included in **Table 2**. We now indicate this by including “(Table 2)” following the sentence that describes this study on page 9 line 143.

- Comment:** P9 line 139: the RCT cited (ref 17) is not in table 3

Response: **Table 3** is correct as is. The in-text citation of ref 17 was an error that we have now removed.

- Comment:** P10 lines 148-151: the ref 36 (RCT) is not in table 3 and the number for ref 37-40 do not match with those reported in the table 2.

Response: We confirm that reference #36 (Ratner 2008), now reference #37 in the revision, is in the second row of **Table 3**. We also ensured that references in text match those reported in **Table 2**.

- Comment:** Line 149: could you precise if dietary pattern was assessed during life course or during pregnancy.

Response: The dietary patterns occurred during mid-life among women with a history of GDM (i.e., a lifecourse exposure rather than a pregnancy-specific exposure). We have articulated this in the text:

page 10 lines 156-158: Two observational studies^{35,36} and one RCT³⁷ assessed associations of healthy dietary patterns during mid-life with risk of incident T2D among women with a history of GDM but showed inconsistent results. Six studies assessed novel biomarkers of T2D risk³⁸⁻⁴³, including metabolomics, lipidomics, sICAM and sE-selectin, and proinsulin-to-insulin ratio.

7. **Comment:** Line 150: is it clinical or biological biomarkers? Can you provide details about the biomarkers and the period when they were explored?

Response: As the biomarkers referenced above are not yet used for standard practice in clinical settings, we removed the term “clinical” and now use the term “novel biomarkers” on page 10 line 158.

Results for Maternal cardiovascular diseases:

8. **Comment:** The references numbers in the text must be checked because some of them don’t match with those in table 4 (for ex, ref 17 does not include maternal BMI)

Response: We have cross-referenced the in-text citations with those in the table to reconcile inconsistencies.

9. **Comment:** Can you report results about BMI during life course in the text? (ref 43).

Response: Former reference #43 (Kaul et al. 2015), now reference #46 in the revised manuscript, explores the association of maternal pre-pregnancy weight status in relation to incident maternal diabetes, hypertension, and CVD. The primary finding is that higher maternal pre-pregnancy weight status based on BMI among women with GDM corresponds with higher risk of T2D, which aligns with all other observational studies that focused on maternal BMI as the exposure. This finding is synthesized in the text:

page 9 lines 139-141: All observational studies that assessed maternal BMI as a prognostic factor showed that higher maternal BMI prior to and/or during pregnancy, and later in the lifecourse predicted higher risk of T2D.

10. **Comment:** For diet and smoking, can you provide details on the period of life for the exposition (pregnancy or life course).

Response: We have done this in the tables. In **Table 1**, the “Exposure” column indicates when the exposures, including lifestyle behaviors like diet and smoking, occurred – i.e., during the perinatal/postpartum or maternal lifecourse. We also have a third category of “social/genetic” for exposures such as race and ethnicity or genetic predisposition that are not constrained to a specific life stage. In tables summarizing observational studies (**Tables 2, 4, 5, and 7**), we either stratify the results by the above exposure categories or indicate the timing of exposures within columns (e.g., “pregnancy diet,” “pre-pregnancy BMI”) if it is not implicit (i.e., “GDM severity” indicates a pregnancy exposure and “breastfeeding” indicates a postnatal/postpartum exposure).

Results for Offspring Anthropometry:

11. **Comment:** P11 line 173: you can add that for the RCT, the association with type of treatment during pregnancy was also considered.

Response: Done.

12. **Comment:** P11 line 181-183: the sentence must be revised.

Response: Thank for catching the extra “and” in the sentence; we have removed it.

Results for Offspring Cardiometabolic profile:

13. **Comment:** P12 line 206: there is only one observational study that is not consistent.

Response: We have modified the wording such that the sentence now reads, “... but this finding was not

recapitulated in other observational studies.” (page 12 lines 221-223)

14. Comment: P12 line 207: in table 8, there is only one RCT (ref 105) that compared diet vs insulin. But, in this study, the intervention was metformin versus insulin, not diet versus insulin.

Response: Thank you for catching this error; we have rectified it in **Table 8**.

Discussion

15. Comment: The paragraph 4.2.2 about maternal T2D should be before the paragraph 4.2.1, in the same order as the results.

Response: Done.

16. Comment: P14 line 241: are at higher risk for CVD

Response: Thank you for identifying the missing “for.” We have included it.

17. Comment P14 line 243: you can add the period of the exposition for maternal BMI (before, during pregnancy or during the life course?)

Response: Most studies explore the women’s pre-pregnancy BMI, which we now include in text on page 15 line 280.

18. Comment: P16 paragraph 4.3.1: there is an issue about the age at follow-up for the offspring (see above). Did you search for adults? If yes and you did not find data about them, you could discuss this point. If not, you should justify this.

Response: As indicated in our response to this a prior comment (Comment #9), this review includes studies of adult offspring. Results of offspring are discussed throughout the paper where relevant.

19. Comment: P17 lines 302-303: you could add that biomarkers should be studied in future studies.

Response: Thank you for the suggestion. We have done this.

Page 17 lines 322-326: This, again, suggests that additional research is needed to better understand the pathophysiology of maternal GDM, to characterize relevant *in utero* programming pathways⁸⁷⁻⁸⁹, and identify accurate and valid prognostic biomarkers – including those in cord blood – as well as outcomes in offspring that are more relevant to future disease risk⁶ such as directly-assessed neonatal adiposity⁸⁹.

20. Comment: P17 line 313: you could underline that you did not find any study about biomarkers in the offspring, particularly biomarkers at birth (cord blood) that could be predictive markers for risk of adverse outcomes.

Response: Thanks, we have done this on page 17 lines 322-326 (as above) and in our concluding sentence on page 19 lines 372-376.

21. Comment: P17 line 320: what do you mean by gold standard assessments?

Response: Gold standard assessments refer to well-accepted methods for research measures. For example, dual x-ray absorptiometry (DXA) is considered gold-standard technique for measuring body composition (c.f., PMID 27398414), along with computed tomography and magnetic resonance imaging (Willet Nutritional Epidemiology, Chapter 10. 2nd Ed. New York: Oxford University Press; 1998:244-272). We now clarify this in text

with the appropriate references (page 17 lines 333-336)

22. Comment: A table synthesising the main exposures identified for each outcomes and the level of evidence will help the reader.

Response: We agree that such a table is helpful and refer the reviewer to **Figures 2-4**, which are visual depictions (heat maps) of the quality of studies based on the Diabetes Canada 2018 Clinical Practice Guidelines.

23. Comment: Table 1: Numbering of the references is right. But, there is 37 references in the table, not 35 references, for offspring outcomes.

Response: Apologies for the discrepancy; we have rectified the error.

24. Comment: Table 2: Eades 2015 (p 43): this paper also reported on postpartum GWG according to the text.

Response: We have re-read the paper and confirm that this study (reference #12) included data on pregnancy weight gain, not postpartum weight gain. Therefore, the information presented in **Table 2** is correct.

25. Comment: Table 2: Tobias 2012 (p 44): can you give precision on the type of diet?

Response: The diets are the alternate Mediterranean diet (aMED), the Dietary Approaches to Stop Hypertension (DASH) diet, and the Alternate Healthy Eating Index (aHEI). We have included this information in **Table 2** for Tobias 2012.

26. Comment: Schaefer-Graf 2009 (p 45): the novel biomarker is in fact a clinical composite score.

Response: This is correct. We now note this in **Table 2**.

27. Comment: Table 4: Pintaudi 2015: can you specify pregnancy complications. It seems it is stillbirth.

Response: Correct, the complication is stillbirth. This is indicated in **Table 4**.

28. Comment: Table 6, 7 and 8: age of outcome assessment should be added.

Response: Done.

29. Comment: Table 8: Battin 2015: the intervention was metformin versus insulin, not diet versus insulin.

Response: Thank you for pointing out this error. The appropriate cell in **Table 8** now reads, "Treatment (Metformin vs. insulin)."

30. Comment: Figure 1: can you add in the flow chart the reasons why studies were excluded at the two first steps. You should also indicate at the end of the flow chart the number of studies included in the review for women and the number for offspring, separately.

Response: Done.

31. Comment: Figure 2, 3 and 4: can you add the number of the references corresponding to each line.

Response: The purpose of the heat maps is to complement our very detailed tables by providing a succinct overview of study quality and illustrate, as we conclude, that current literature on prognostic biomarkers and risk factors for cardiometabolic disease among women and children affected by GDM pregnancies is low quality and, accordingly, additional research is needed to advance precision prognostics of GDM. We also note that the

heat maps include important details such as the lead author's surname, year of publication, cohort/study name, and exposures + outcomes of interest. Therefore, unless the Editor feels strongly otherwise, we prefer to keep these figures as they are.

32. Comment: Figure 3: I understood that the first line refers to ref 85 and 76 and then it also includes offspring outcomes.

Response: The first line of **Figure 3** refers to Ref 85 (**now reference #95:** Casey et al. *Am J Perinatal* 2020; 37(5):475; PMID 30866027), which focuses on maternal outcomes only and is classified as such in the figure. Ref 76 (**now reference #75:** Landon et al. *Diabetes Care* 2015;38(3):445; PMID 25414152) focuses on child outcomes and is referenced in the 3rd line under the "Offspring outcomes" portion of the figure. Both studies use data from the Eunice Kennedy Shriver National Institute of Child Health and Human Development MFMU Network's Multi-Center RCT, but the outcomes of interest are among women only for Casey et al. and among offspring only for Landon et al., as shown in **Figure 3**.

Reviewer #2:

- Comment:** This compilation of the evidence on GDM and progression to type 2 diabetes, cardiometabolic outcomes and risk of cardiovascular disease provides an important contribution to the literature. The assembly of the publications is well-organized and mostly reasonable. However, the authors have omitted several studies that make important contributions to the literature, and have the highest quality of design. For example, the rigorous systematic repeated testing of cardiometabolic profiles in both SWIFT and CARDIA are rare. In addition, CARDIA contributes research measures of cardiovascular disease risk and full glucose tolerance many years after pregnancy. Several publications from these diverse prospective USA research cohorts, the SWIFT Study and the CARDIA Study have been omitted from the review. The **CARDIA** Study published several studies on GDM and cardiovascular health outcomes, including how glucose tolerance may affect cIMT and CAC measures of CVD risk. The **SWIFT** Study includes: SWIFT Women Study (n=1033), the SWIFT Offspring Study of infants (n=550), and the SWIFT Study in Youth (n=1033) children. A key study on early diet in SWIFT children was omitted and is attached below. In SWIFT Women, there are also several publications on metabolomics and lipidomics from SWIFT were also omitted from this review. The list of these studies and their features (summarized below), as well as the supplemental files and a manuscript are attached. For a comprehensive evaluation of the studies, it would be important to state identify studies that obtained direct measures of "GDM severity" from the EHR, and that most studies of GDM offspring simply classify individuals as GDM "yes or no". Also, the CARDIA study measured glucose tolerance before pregnancies, and thus, is one of the only studies that could rule out diabetes before pregnancy. This feature should be emphasized in the review. Gunderson et al. AJE 2010 analyzed fasting cardiometabolic parameters that predicted GDM. The standardized preconception measures of risk factors is a unique strength of CARDIA. A recent review paper shown below cites the relevant manuscripts published on GDM, T2D and CVD outcomes from CARDIA: Kim C, Catov J, Schreiner PJ, Appiah D, Wellons MF, Siscovick D, Calderon-Margalit R, Huddlestone H, Ebong IA, Lewis CE. Women's Reproductive Milestones and Cardiovascular Disease Risk: A Review of Reports and Opportunities From the CARDIA Study. J Am Heart Assoc. 2023 Mar 7;12(5):e028132. doi: 10.1161/JAHA.122.028132. Epub 2023 Feb 27. PMID: 36847077; PMCID: PMC10111436.

Response: We are grateful to the reviewer for their time and suggestions for additional studies to include. We have responded to details regarding suggested publications, below.

Details about the publications to include:

- Comment:** Findings from the CARDIA have been omitted from Table 2 that shows the prognostic factors associated with type 2 diabetes. The following paper includes a table showing the independent associations between maternal BMI, lifestyle behaviors, race disparities and other factors related to type 2 diabetes progression -- lactation was the main effect, but we reported other factors that influence progression. I attach the documents with the Relative Hazards for the factors above. See Supplemental Figure 3 attached pdf.
 - Reference: Gunderson EP, Lewis CE, Lin Y, Sorel M, Gross M, Sidney S, Jacobs DR Jr, Shikany JM, Quesenberry CP Jr. Lactation Duration and Progression to Diabetes in Women Across the Childbearing Years: The 30-Year CARDIA Study. JAMA Intern Med. 2018 Mar 1;178(3):328-337. doi: 10.1001/jamainternmed.2017.7978. PMID: 29340577; PMCID: PMC5885916.
- Response:** We reviewed the results and the supplementary figure regarding this article, which includes both mothers with and without GDM. Since we cannot evaluate the associations of interest exclusively among women with GDM in the supplemental figure, we elect not to include these additional data in the table.
- Comment:** Several studies have been omitted from Table 4 on the prognostic factors for cardiovascular disease among women with GDM.
 - This paper evaluates the impact of progression to normoglycemia, prediabetes and diabetes after GDM and non-GDM associated with risk of Coronary Artery Calcium, a strong risk factor for CVD in young adults: Gunderson EP, Sun B, Catov JM, Carnethon M, Lewis CE, Allen NB, Sidney S, Wellons M, Rana JS, Hou L, Carr JJ. Gestational Diabetes History and Glucose Tolerance After Pregnancy Associated With Coronary Artery Calcium in Women During Midlife: The CARDIA Study. Circulation. 2021 Mar

9;143(10):974-987. doi: 10.1161/CIRCULATIONAHA.120.047320. Epub 2021 Feb 1. PMID: 33517667; PMCID: PMC7940578.

- This paper evaluates the relationship between GDM and no GDM with and without prediabetes and CVD risk factors related to carotid artery IMT: Gunderson EP, Chiang V, Pletcher MJ, Jacobs DR, Quesenberry CP, Sidney S, Lewis CE. History of gestational diabetes mellitus and future risk of atherosclerosis in mid-life: the Coronary Artery Risk Development in Young Adults study. *J Am Heart Assoc.* 2014 Mar 12;3(2):e000490. doi: 10.1161/JAHA.113.000490. PMID: 24622610; PMCID: PMC4187501.
- This paper evaluates changes in cardiometabolic risk factors related to GDM: Catov JM, Sun B, Bertolet M, Snyder GG, Lewis CE, Allen NB, Shikany JM, Ingram KH, Appiah D, Gunderson EP. Changes in Cardiometabolic Risk Factors Before and After Gestational Diabetes: A Prospective Life-Course Analysis in CARDIA Women. *Obesity (Silver Spring).* 2020 Aug;28(8):1397-1404. doi: 10.1002/oby.22848. Epub 2020 Jul 6. PMID: 32627948; PMCID: PMC7501158.

Response: We thank the reviewer for noting these papers. Per our criterion of including papers among women with GDM for which a clinical outcome of CVD or T2D was assessed, the above studies were not included because none of the outcomes were hard clinical endpoints – i.e., coronary artery calcification (PMID 33517667) and carotid intima media thickness (PMID 24622610) are not disease outcomes but rather, biomarkers of CVD pathogenesis; for PMID 32627948, the outcomes comprised changes in weight and waist circumference following a GDM vs. non-GDM pregnancy, neither of which are clinical T2D or CVD endpoints.

4. **Comment:** The review also omitted an important prospective, longitudinal study from the SWIFT Study in Children (n=844) This longitudinal study from pregnancy, birth and up to age 2-5 years evaluated the impact of maternal risk factors, GDM severity and detailed early infant diet (sugar sweetened beverages, natural fruit juice) from birth to age 1 years and increase subsequent risk of child obesity at 2-5 years. This SWIFT study is a high diverse cohort, and includes with 25% low income families. It is much stronger than the WIC cross-sectional study of children that had diet and BMI assessed at a single time point. This is an important analysis, and one of the only studies published that evaluated infant diet and breastfeeding, specifically sugary beverages, and fruit juice from birth to 1 year.
- Reference: Vandyousefi S, Davis JN, Gunderson EP. Association of infant diet with subsequent obesity at 2-5 years among children exposed to gestational diabetes: the SWIFT study. *Diabetologia.* 2021 May;64(5):1121-1132. doi: 10.1007/s00125-020-05379-y. Epub 2021 Jan 26. PMID: 33495846; PMCID: PMC8016720 -- detailed prospectively monthly surveys to assess infant feeding -- milk and all dietary intake of foods and beverages. Follow up with EHR measurements of child BMI.)

Response: Thank you for bringing this article to our attention. We have included this paper to our review, describe findings in the manuscript (page 12, lines 193-196), and incorporated it into the tables and figures accordingly.

5. **Comment:** There are also several manuscripts from the SWIFT Study on the prediction of type 2 diabetes in a prospective cohort study of 1010 women with GDM who were tested with 2 h 75 g OGTT from 6-9 weeks postpartum and annually tested for progression to type 2 diabetes. We have conducted several nested case control studies (largest # cases of incident diabetes) in a single prospective GDM cohort in follow up from early postpartum. Only one of these studies was cited. We list a few of the other key studies that have been innovative for metabolomic and lipidomic studies of pathways from GDM to future type 2 diabetes. These are important studies.
- Allalou A, Nalla A, Prentice KJ, Liu Y, Zhang M, Dai FF, Ning X, Osborne LR, Cox BJ, Gunderson EP, Wheeler MB. A Predictive Metabolic Signature for the Transition From Gestational Diabetes Mellitus to Type 2 Diabetes. *Diabetes.* 2016 Sep;65(9):2529-39. doi: 10.2337/db15-1720. Epub 2016 Jun 23. PMID: 27338739; PMCID: PMC5001181.
 - Lai M, Al Rijjal D, Röst HL, Dai FF, Gunderson EP, Wheeler MB. Underlying dyslipidemia postpartum in women with a recent GDM pregnancy who develop type 2 diabetes. *Elife.* 2020 Aug 4;9:e59153. doi: 10.7554/eLife.59153. PMID: 32748787; PMCID: PMC7417169.

- C. Lai M, Liu Y, Ronnett GV, Wu A, Cox BJ, Dai FF, Röst HL, Gunderson EP, Wheeler MB. Amino acid and lipid metabolism in post-gestational diabetes and progression to type 2 diabetes: A metabolic profiling study. *PLoS Med.* 2020 May 20;17(5):e1003112. doi: 10.1371/journal.pmed.1003112. PMID: 32433647; PMCID: PMC7239388.

Response: We appreciate the reviewer for pointing out these papers. Lai et al. *PLoS Med* 2020 was already included in this review. We now also include Allalou et al. *Diabetes* 2016 and Lai et al. *eLife* 2020.

Reviewers' comments:

Reviewer #1 (Remarks to the Author):

I thank the authors for the work done. Most of my concerns have been addressed. Anyway, I have still minor comments :

I have not check all the correspondence between the numbering of the references in the main text and the tables, but I have found some mistakes.

For example, for race and ethnicity for maternal T2D (p 9_10, lines 148-150), the authors say there are 4 observational studies, but finally there are 5 references cited (14 to 18) and ref 18 is not in table 2.

Other example, for biomarkers (p10, lines 158-159), six studies are cited in the text (ref 38-43), but I can only find four in table 2 where the numbers of the ref are 13, 30, 103, 104.

The authors should once more carefully check the number of references and the correspondence between the main text and the tables.

Comment 27: "stillbirth" was not added in table 4 for ref Pinaudi 2015.

Comment 30: I have suggested that the number of studies included in the review for women and the number for offspring, to be indicated separately at the end of the flow chart. It's not done.

Reviewer #3 (Remarks to the Author):

I think the comments from reviewer 2 are addressed as these are mainly focusing on addition of suggested studies only.

Reviewers' comments:

Reviewer #1

1. **Comment:** I thank the authors for the work done. Most of my concerns have been addressed. Anyway, I have still minor comments.

Response: We are grateful to the reviewer for their attention to detail in vetting this paper.

2. **Comment:** I have not check all the correspondence between the numbering of the references in the main text and the tables, but I have found some mistakes. For example, for race and ethnicity for maternal T2D (p 9-10, lines 148-150), the authors say there are 4 observational studies, but finally there are 5 references cited (14 to 18) and ref 18 is not in table 2.

Response: Thank you for catching this error. We have modified the above-mentioned section, including removal of reference #18 (see page 10 line 150). Additionally, we have thoroughly reviewed the in-text citations with the tables and final references to ensure that the correct papers were noted.

3. **Comment:** Other example, for biomarkers (p10, lines 158-159), six studies are cited in the text (ref 38-43), but I can only find four in table 2 where the numbers of the ref are 13, 30, 103, 104. The authors should once more carefully check the number of references and the correspondence between the main text and the tables.

Response: Thank you for identifying this oversight. We confirm that the original references 38-43 are in **Table 2**, and now include refs 13, 30, 103, and 104 in the in-text citation, in addition to refs 38-43. For completeness, we list the references below, though the numeric values have changed in the revision. We note the new reference numbers at the end of each citation, below. The revised sentence and updated in-text citations are on page 10 lines 158-159.

Original references #38-43

38. Tobias DK, Clish C, Mora S, et al. *Clinical chemistry*. Dietary Intakes and Circulating Concentrations of Branched-Chain Amino Acids in Relation to Incident Type 2 Diabetes Risk Among High-Risk Women with a History of Gestational Diabetes Mellitus. 64(8):1203-1210. 2018. (**now reference #37**)
39. Lai M, Liu Y, Ronnett GV, et al. *PLoS medicine*. Amino acid and lipid metabolism in post-gestational diabetes and progression to type 2 diabetes: A metabolic profiling study. 17(5):e1003112. 2020. . (**now reference #38**)
40. Göbl CS, Bozkurt L, Yarragudi R, et al. *Cardiovascular diabetology*. Biomarkers of endothelial dysfunction in relation to impaired carbohydrate metabolism following pregnancy with gestational diabetes mellitus. 13:138. 2014. (**now reference #39**)
41. Hanson U, Persson B, Hartling SG, Binder C. *Diabetes care*. Increased molar proinsulin-to-insulin ratio in women with previous gestational diabetes does not predict later impairment of glucose tolerance. 19(1):17-20. 1996. (**now reference #40**)

42. Lai M, Al Rijjal D, Röst HL, Dai FF, Gunderson EP, Wheeler MB. *eLife*. Underlying dyslipidemia postpartum in women with a recent GDM pregnancy who develop type 2 diabetes. 2020. (**now reference #41**)
43. Allalou A, Nalla A, Prentice KJ, et al. *Diabetes*. A Predictive Metabolic Signature for the Transition From Gestational Diabetes Mellitus to Type 2 Diabetes. 65(9):2529-2539. 2016. (**now reference #42**)

References that we added to the in-text citations, above, in response to the reviewer's comment:

13. Schaefer-Graf UM, Klavehn S, Hartmann R, et al. *Diabetes care*. How do we reduce the number of cases of missed postpartum diabetes in women with recent gestational diabetes mellitus? 32(11):1960-1964. 2009. (**still reference #13**)
30. Löbner K, Knopff A, Baumgarten A, et al. *Diabetes*. Predictors of postpartum diabetes in women with gestational diabetes mellitus. 55(3):792-797. 2006. (**now reference #29**)
103. Lappas M, Mundra PA, Wong G, et al. *Diabetologia*. The prediction of type 2 diabetes in women with previous gestational diabetes mellitus using lipidomics. 58(7):1436-1442. 2015. (**now reference #43**)
104. Weerasiri T, Riley SF, Sheedy MT, Walstab JE, Wein P. *The Australian & New Zealand journal of obstetrics & gynaecology*. Amniotic fluid insulin values in women with gestational diabetes as a predictor of emerging diabetes mellitus. 33(4):358-361. 1993. . (**now reference #44**)

4. **Comment:** Comment 27: "stillbirth" was not added in table 4 for ref Pinaudi 2015.

Response: We confirm that "stillbirth" is included in parentheses after "Delivery complication" for Pinaudi 2015 in **Table 4**.

5. **Comment:** Comment 30: I have suggested that the number of studies included in the review for women and the number for offspring, to be indicated separately at the end of the flow chart. It's not done.

Response: We thank the reviewer for this suggestion. The structure of the flow charts were standardized across the multiple reviews conducted under the broader ADA/EASD PMDI effort; however, we have added the total number of studies with offspring outcomes and maternal outcomes to the Consort figure.

Reviewer #3

1. **Comment:** I think the comments from reviewer 2 are addressed as these are mainly focusing on addition of suggested studies only.

Response: Thank you.

REVIEWERS' COMMENTS:

Reviewer #1 (Remarks to the Author):

All my comments were addressed. Thanks to the authors.